# ⚖️DiPlomat:
# A Dialogue Dataset for Situated Pragmatic Reasoning

**Hengli Li**[1,2]
lihengli@stu.pku.edu.cn

**Song-Chun Zhu**[1,3,4,5]
s.c.zhu@pku.edu.cn

**Zilong Zheng**[1✉]
zlzheng@bigai.ai

[1] National Key Laboratory of General Artificial Intelligence, BIGAI
[2] Yuanpei College, PKU  [3] Institute for Artificial Intelligence, PKU
[4] School of Intelligence Science and Technology, PKU  [5] Department of Automation, THU
✉ Corresponding author.

https://diplomat-dataset.github.io

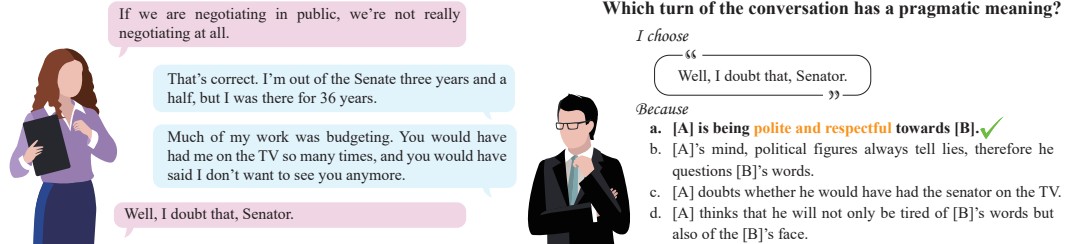

Figure 1: Illustration of **DiPlomat** dataset. Left: Example of a pragmatic conversation. Right: Pragmatic Identification and Reasoning task.

## Abstract

The ability to discern and comprehend pragmatic meanings is a cornerstone of social and emotional intelligence, referred to as pragmatic reasoning. Despite the strides made in the development of Large Language Models (LLMs), such as Chat-GPT, these models grapple with capturing the nuanced and ambiguous facets of language, falling short of the aspiration to build human-like conversational agents. In this work, we introduce a novel benchmark, the **DiPlomat**, which delves into the fundamental components of conversational pragmatic reasoning, encompassing situational context reasoning, open-world knowledge acquisition, and unified figurative language understanding. We start by collecting a new human-annotated dialogue dataset, composed of 4,177 multi-turn dialogues and a vocabulary of 48,900 words. Along with the dataset, two tasks are proposed to evaluate machines' pragmatic reasoning capabilities, namely, Pragmatic Reasoning and Identification(PIR) and Conversational Question Answering (CQA). Furthermore, we probe into a zero-shot natural language inference task, where the significance of context in pragmatic reasoning is underscored. Experimental findings illustrate the existing limitations of current prevailing LLMs in the realm of pragmatic reasoning, shedding light on the pressing need for further research to facilitate the emergence of emotional intelligence within human-like conversational agents.

37th Conference on Neural Information Processing Systems (NeurIPS 2023) Track on Datasets and Benchmarks.

# 1 Introduction

The fabric of human sociality is made up of complicated relations that evolve through different dimensions of interaction and communication channels [1, 2]. Social consensuses, such as social norms and values, are thereby formed between humans that convey meanings of individual minds, including beliefs, intentions and desires [3]. In a process of effective negotiation and social conversation, particularly, such social behaviors are only partly driven by **literal** meanings that are *objective*, *rational* and *explicit* [4, 5]. Instead, these behaviors are commonly governed by affective or **pragmatic** meanings of dialogue utterances that refer to the emotional and cultural meanings of conversational partners and are *subjective*, *emotional* and *implicit*. For instance in Fig. 1, the lady responds with "I doubt that" rather than "I am not tired of you" to express a sense of humor and politeness. The competency of perceiving such *pragmatic* meanings is crucial to social and emotional intelligence (EI) [6] and is referred to as **pragmatic reasoning**.

The rapid developments of large language models (LLMs), such as ChatGPT and InstructGPT [7], have set off a wave of the next generation of conversational AI over the recent years. Despite the inspiring capabilities of language generation [8] and reasoning [9] achieved with massive computational resources and tremendous natural language data, LLMs barely show convincing communicative skills [10], *i.e.*, they fail to capture pragmatic and ambiguous meanings of input prompts [11, 10, 12]. Critically, current neural generative models are trained to be objective with safe and satisfiable responses [13, 14], which largely deviates from the long-standing goal of building a human-like agent. Recently, Meta Research Team *et al.* [5] introduce a ChatBot that demonstrates human-level play in a language board game *Diplomacy* where lying and misleading commonly occur. However, their main focus is on game policy learning rather than pragmatic reasoning.

**What are the core components of real-life conversational pragmatic reasoning?** Motivated by theories of cognitive linguistics [1–3] and conversational modeling [4, 5, 15, 16], we anticipate it to be three-fold:

- *Situational Context* Reasoning. Understanding pragmatic meaning requires a detailed understanding of conversational contexts. Consider the utterance "You are making the rest of us looking bad", under different situations of praise and sarcasm, the sentence may convey completely opposite meanings. Furthermore, typical conversational reasoning challenges such as coreference resolution and intention prediction are largely dependent on the success of situated context modeling.

- *Open-world Knowledge* Acquisition. The open-world knowledge includes commonsense knowledge (*e.g.*, social ethics) that can be learned from different domains of dialogue corpus and domain-specific knowledge (*e.g.*, American histories). Successful pragmatic reasoning requires the acquisition of open-world knowledge and joint reasoning over the conversation.

- Unified *figurative language* understanding. Figurative language is one of the most frequently used tricks for conveying implicit meanings with subjective emotions. Previous works treat different forms of figurative language understanding as individual tasks, such as metaphors [17], idioms [18, 19], pun [20], *etc*. Pragmatic reasoning provides a feasible unified perspective that considers all these tasks as recovering their literal meanings.

In order to step towards a general human-like communicative agent, in this work, we introduce **DiPlomat**, a real-life **Di**alogue dataset that focuses on **Prag**matic reasoning. **DiPlomat** stems from an interview dataset [21], and experiences three steps of curation: automatic selection, fine-grained manual annotation and human refinement (Sec. 3). Our dataset comprises 4, 177 dialogues and covers a vocabulary of 48, 900 words. More than that, human-annotated answers reach the amount of 6, 494 and hold a vocabulary size of 20, 000. Along with the dataset, we propose two tasks, Pragmatic Identification and Reasoning (PIR) and Conversational Question Answering (CQA), to benchmark machines' pragmatic reasoning capabilities (Sec. 4). The CQA task possesses 19, 482 questions concerning the content of collected dialogues and the answers to the questions are written by humans. We run extensive experiments on previous state-of-the-art models on **DiPlomat** (Sec. 5). The best model achieves less than 0.70 accuracy score in PIR, and none of the models achieve more than 0.50 accuracy score for CQA. Moreover, we test previous pre-trained LLMs' (including ChatGPT) zero-shot reasoning capability with a natural language inference (NLI) task. Regarding the experimental results provided, the significance of pragmatic reasoning speaks for itself. Throughout a thorough analysis of the limitations of current models, we aim to shed light on future research toward building general conversational agents.

Table 1: Comparisons on similar datasets and our dataset. QA: Question Answering. NUP: Next Utterance Prediction. NLI: Natural Language Inference. PI: Plausible Inference. IR: Implicature Recovery. PIR: Pragmatic Identification and Reasoning. Manually: the dataset is being checked or collected by humans. Diverse: the dataset lies in several specific domains. Open: the dataset doens't fall into particular domains.

| Dataset | Domain | Manually | Task | Implicature | Reasoning | Multi-Turn |
|---|---|---|---|---|---|---|
| Ubuntu [ACL 2015] [22] | Technique | ✗ | NUP | ✗ | ✗ | ✓ |
| RACE [EMNLP 2017] [23] | Open | ✗ | QA | ✗ | ✓ | ✗ |
| ARC [ArXiv 2018] [24] | Science | ✗ | QA | ✗ | ✓ | ✗ |
| MNLI [NAACL 2018] [25] | Open | ✓ | NLI | ✗ | ✓ | ✗ |
| Persona-Chat [ACL 2018] [26] | Persona | ✓ | NUP | ✗ | ✗ | ✓ |
| SWAG [EMNLP 2018] [27] | Movie | ✗ | PI | ✗ | ✓ | ✗ |
| Cosmos QA [EMNLP 2019] [28] | Persona | ✓ | QA | ✗ | ✓ | ✗ |
| CoQA [NAACL 2019] [29] | Diverse | ✓ | QA | ✗ | ✓ | ✓ |
| DREAM [ACL 2019] [30] | Open | ✓ | QA | ✗ | ✓ | ✓ |
| Dialogue NLI [ACL 2019] [31] | Persona | ✗ | NLI | ✗ | ✗ | ✓ |
| DROP [ACL 2019] [32] | Open | ✗ | QA | ✗ | ✓ | ✗ |
| MuTual [ACL 2020] [33] | Open | ✓ | NUP | ✗ | ✓ | ✓ |
| IMPPRES [ACL 2020] [34] | Open | ✗ | NLI | ✓ | ✓ | ✗ |
| GRICE [ACL 2021] [12] | Daily | ✗ | IR & QA | ✓ | ✓ | ✓ |
| **DiPlomat** | Open | ✓ | PIR & QA | ✓ | ✓ | ✓ |

## 2 Related Work

**Conversational Dataset** Tab. 1 provides a comparative analysis of our dataset with similar conversational datasets. The Dream [30] dataset formalizes dialogues from English exams into question-answering task with a focus on in-depth dialogue comprehension. With question-answer pairs compiled by two annotators, CoQA [29] focuses on reasoning in conversation understanding. By utilizing English listening comprehension tests, MuTual [33] is built to address the issue of general dialogue reasoning. In contrast to these preceding datasets, the GRICE [12] dataset represents a significant advancement in the field of pragmatic reasoning as it incorporates implicature and reasoning. However, both MuTual and GRICE exhibit a shared limitation in that they do not possess data that closely resembles real-world interactions, leading to a lack of diversity in their respective datasets. Additionally, other relevant datasets, including commonsense [28], reasoning [23], and natural language inference (NLI) [31, 25], are also included in Tab. 1. By examining various dimensions such as domain, manual annotation, task variety, implicature, reasoning, and multi-turn interactions, our dataset offers unique advantages in addressing the challenge of pragmatic reasoning in dialogues.

**Language Models for Dialogue Generation** Conversational AI has emerged as a prominent research area within the field of natural language processing (NLP), attracting significant attention and interest. Numerous pre-trained models have been proposed to tackle dialogue generation tasks, such as DialogGPT [35], GODEL [36], LaMDA [37], and Meena [38], and they have achieved marvelous results on competitions [39]. Furthermore, a pivotal milestone has been achieved with the advent of ChatGPT, garnering widespread interest and stimulating further investigation in the domain of conversational AI. ChatGPT, built upon the principles of transformer models [40], undergoes training on an extensive corpus of data, resulting in its profound efficacy. Notably, this system boasts an impressive magnitude of billions of parameters. In a notable study conducted by OpenAI [13], it has been demonstrated that the augmentation of parameters, referred to as the Scaling Law, substantially enhances the model's capabilities. Also, the enlargement of the number of parameters triggers the emergence of miraculous ability. After the success of ChatGPT, more models such as PaLM 2 [41] appeared in the field.

**Pragmatic Reasoning** Pragmatic reasoning is a significant subject within the field of pragmatics, attracting considerable attention from linguists. The Gricean maxims, which is one of the most important achievements, serves as a foundational theory within the domain of pragmatics. This theoretical framework comprises four distinct maxims: (1) The Maxim of Quality, (2) The Maxim of Quantity, (3) The Maxim of Relevance, and (4) The Maxim of Manner [15, 16]. In contrast to rigid

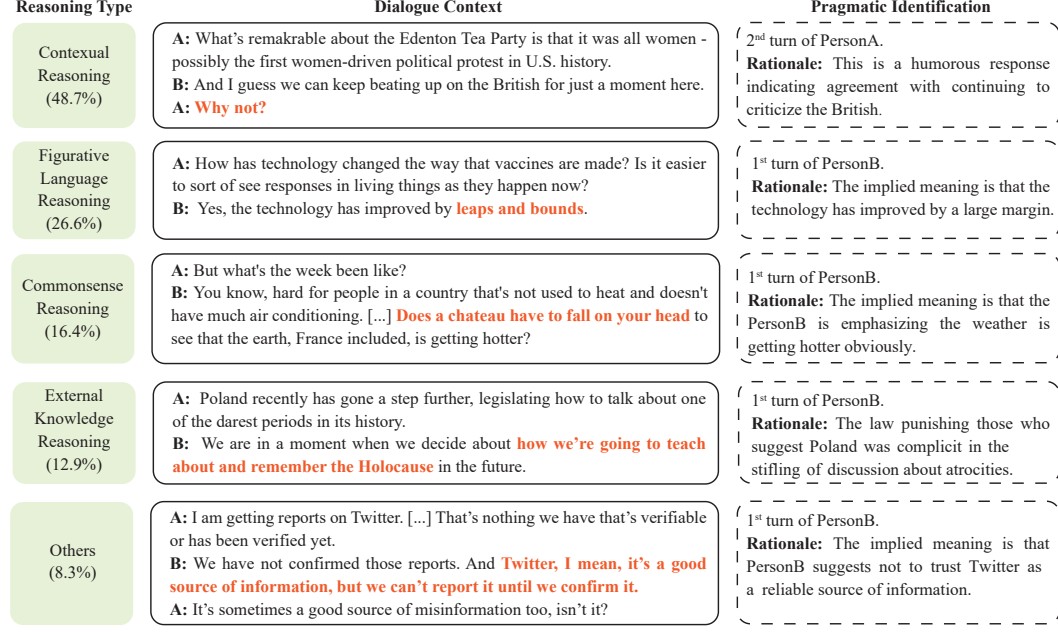

| Reasoning Type | Dialogue Context | Pragmatic Identification |
|---|---|---|
| Contextual Reasoning (48.7%) | **A:** What's remarkable about the Edenton Tea Party is that it was all women - possibly the first women-driven political protest in U.S. history. 
 **B:** And I guess we can keep beating up on the British for just a moment here. 
 **A: Why not?** | 2nd turn of PersonA. 
 **Rationale:** This is a humorous response indicating agreement with continuing to criticize the British. |
| Figurative Language Reasoning (26.6%) | **A:** How has technology changed the way that vaccines are made? Is it easier to sort of see responses in living things as they happen now? 
 **B:** Yes, the technology has improved by **leaps and bounds**. | 1st turn of PersonB. 
 **Rationale:** The implied meaning is that the technology has improved by a large margin. |
| Commonsense Reasoning (16.4%) | **A:** But what's the week been like? 
 **B:** You know, hard for people in a country that's not used to heat and doesn't have much air conditioning. [...] **Does a chateau have to fall on your head** to see that the earth, France included, is getting hotter? | 1st turn of PersonB. 
 **Rationale:** The implied meaning is that the PersonB is emphasizing the weather is getting hotter obviously. |
| External Knowledge Reasoning (12.9%) | **A:** Poland recently has gone a step further, legislating how to talk about one of the darest periods in its history. 
 **B:** We are in a moment when we decide about **how we're going to teach about and remember the Holocause** in the future. | 1st turn of PersonB. 
 **Rationale:** The law punishing those who suggest Poland was complicit in the stifling of discussion about atrocities. |
| Others (8.3%) | **A:** I am getting reports on Twitter. [...] That's nothing we have that's verifiable or has been verified yet. 
 **B:** We have not confirmed those reports. And **Twitter, I mean, it's a good source of information, but we can't report it until we confirm it.** 
 **A:** It's sometimes a good source of misinformation too, isn't it? | 1st turn of PersonB. 
 **Rationale:** The implied meaning is that PersonB suggests not to trust Twitter as a reliable source of information. |

Figure 2: **DiPlomat** dataset samples. Each row illustrates an exemplar case with its reasoning type, dialogue context, pragmatic turn, and the corresponding rationale. Evidence that support the pragmatic identification are marked in orange.

rules or theorems, the Gricean maxims, which capture the prevalent dynamics of conversations, are susceptible to frequent breaches in the context of human communication. These breaches, stemming from the intricacies of real-world interaction, notably manifest in the violation of one or more of these maxims. Such breaches, aligned with the cooperative principle, give rise to pragmatic phenomena that necessitate the engagement of pragmatic reasoning by recipients of the communication [15]. In the field of natural language processing, previous work tries to model the problem, but predominantly focus on specific types of phenomena. For example, EPIE [18], PIE [19] center around idiomatic expressions, while MOVER [42] emphasizes hyperbole, and MERMAID [17] investigates metaphor usage. However, paronomasia is much more under-studied [20], with researchers frequently intertwining it with humor [43]. In a related vein, GRICE [12] endeavors to study implicature in a unified manner, but its data does not originate from real-world contexts, thereby lacking diversity and exhibiting conspicuous patterns. Insights into the comprehension of figurative language are provided by Stowe et al. [44] and Chakrabarty et al. [45], with the former specifically delving into the realm of metaphors and idioms, and the latter investigating idioms and similes. The successful completion of our task needs the incorporation of commonsense knowledge. Extensive scholarly efforts have been dedicated to addressing the challenge of leveraging commonsense in various works [33, 46–48].

## 3 The DiPlomat Dataset

### 3.1 Data Source Construction

The **DiPlomat** dataset stems from the Interview dataset [21], which consists of two subsets, a two-party subset comprising 23,714 dialogues and a multi-party subset with 105,848 dialogues. Given our specific focus on conversations involving only two communicators, we exclusively utilize the two-party subset for our research purposes. The Interview dataset itself is a real-world collection of NPR radio transcripts, spanning a period of 20 years of NPR programs. The curation process for our dataset involved several stages, including automatic selection, fine-grained annotation, and human refinement. Details are provided as follows.

**Step I. Automatic Selection.**    The extensive size of the source dataset introduces redundancy, and thus requires automatic measures to alleviate the burden of human annotation. Therefore, we employ algorithms and models to perform an initial filtering process. In order to establish a unified framework,

we consider various types of pragmatic phenomena and utilize different techniques to extract relevant instances from the source dataset. For instance, we utilize the EPIE list [18] for a string-matching method to identify idioms in dialogues, and we train RoBERTa [49] on Hypo-XL [50] for hyperbole detection; refer to Appendix B.

**Topic Segmentation**  Topic segmentation is a small operation taken after automatic selection. The original dialogues employed in our study consist of lengthy and multi-turn exchanges, which are ill-suited for our research objectives. Consequently, we implement a segmentation process to break down these dialogues into shorter units. To achieve this, we employ two techniques, namely BERTScore [51] and TextTiling [52]. The segmentation procedure starts with computing the BERTScore between adjacent turns and subsequently applying the TextTiling algorithm to the generated BERTScores.

**Step II. Fine-grained Annotation.**  We leverage Amazon Mechanical Turk (AMT) to conduct detailed annotation of pragmatic turns within our dialogues. Workers participating in the annotation task are instructed to select all turns that exhibit a divergence between their literal meaning and their intended meaning. Due to the subjective nature of pragmatic reasoning, we request the workers to provide confidence scores along with reasons for their choices and for each dialogue two workers are recruited. All annotators shall meet the following criteria: (i) from English-speaking countries; (ii) Completion of a minimum of 1,300 tasks with a 98% approval rate. We also present them with detailed instructions and four examples so that workers are clear about our objectives and requirements. The instructions part outlines the step-by-step procedures for accomplishing the assigned tasks and highlights some key points that workers should pay attention to. The four examples offered are representative of classical pragmatic conversations drawn from the field of linguistics, serving as practical references for the workers. To mitigate the intricacies arising from the identities of dialogue communicators, a simplified representation is adopted, whereby the speakers are denoted as `PersonA` and `PersonB`. To ensure the reasonableness and quality of the data, we manually examined 30 data samples and blocked workers who are unqualified. As a result, a total of 5,869 dialogues are selected; refer to Appendix B for details.

**Step III. Human Refinement.**  In this process, tasks for workers are formulated as multiple-choice questions. Previously collected human-annotated reasons are transformed into choices, utilizing a template format: `[turn {turn_id}: {reason}]`. In addition, to mitigate the impact of careless workers, we introduce a distractor choice for each gold choice. These distractor choices are generated using BERTScore [51] by selecting the reason with the highest score from other unrelated dialogues. Of note, for each dialogue, an equal number of gold choices and distractor choices are provided. Workers are requested to select all reasonable choices for each conversation and are warned of the presence of distractor choices. Workers who frequently select distractor choices are blocked, and their work is rejected. Through this refinement process, 1,692 dialogues are filtered out, while 4,177 dialogues are preserved, ensuring the integrity and reliability of our dataset; refer to Appendix B for more details.

Table 2: Statistical feature of **DiPlomat**.

| | |
|---|---|
| **# Dialogues** | $4.17 \times 10^3$ |
| **Avg. Turns per Dialogue** | 4.10 |
| **Avg. Words per Turn** | 42.80 |
| **Avg. Human Reason per Dialogue** | 1.56 |
| **Avg. Words per Human Annotated Reason** | 25.31 |
| **Vocabulary Size (dialogue)** | $4.89 \times 10^4$ |
| **Vocabulary Size (human-annotated reasons)** | $2.00 \times 10^4$ |

### 3.2 Dataset Statistics

The **DiPlomat** dataset comprises a total of 4,177 multi-turn dialogues, with each dialogue averaging 4.1 turns. On average, there are 1.56 pragmatic turns per dialogue. The distribution of dialogues with different quantities of pragmatic turns is illustrated in Fig. 4; see Tab. 2 for detailed dataset statistics.

With respect to the motivation introduced in Sec. 1, we categorize the process of transitioning from dialogue to human-annotated rationales into five reasoning types:

- **Contextual Reasoning:** The comprehension of the context is paramount for this reasoning process.
- **Figurative Language Reasoning:** Proficiency in understanding figurative language, such as idioms and metaphors, is indispensable for advancing this type of reasoning.
- **Commonsense Reasoning:** The utilization of commonsense knowledge, such as recognizing that a chateau cannot fall from the sky, is vital for this category.

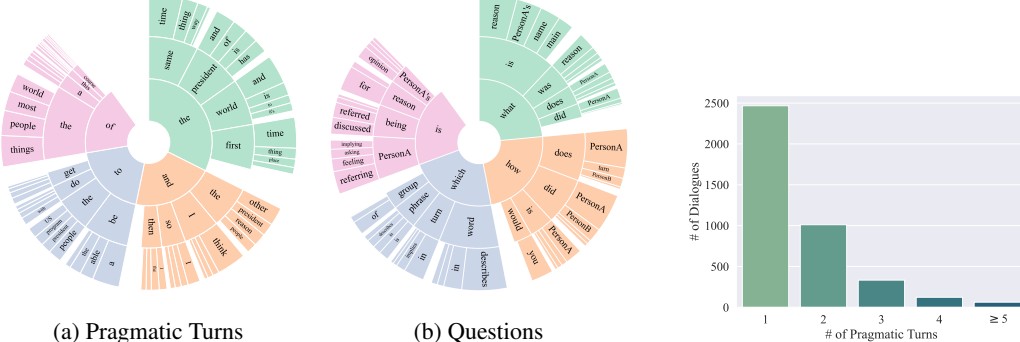

|                  |                  |
| :--------------: | :--------------: |
| (a) Pragmatic Turns | (b) Questions |

Figure 3: Distribution of trigram words of pragmatic turns and questions.

Figure 4: Distribution of dialogues *w.r.t.* # of pragmatic turns.

- **External Knowledge Reasoning:** This form of reasoning necessitates knowledge that extends beyond commonsense and is not explicitly mentioned in the dialogue.
- **Others:** This category includes pragmatic dialogues that fit none of the above.

Fig. 2 demonstrates the proportion of each type. The prevalence of data within the context partition prove the significance of context in pragmatic reasoning of real life. Fig. 3a depicts a sunburst visualization illustrating the distribution of trigram words within pragmatic turns. The diverse range of trigram words indicates that the **DiPlomat** dataset enjoys the rich diversity from real-life corpora, and covers a wide array of topics. In addition, the recurring occurrence of the words "president'' and "world'' is observed, demonstrating **DiPlomat**'s slight bias to politics and world-wide events.

## 4   Task Definition

We propose 2 distinct tasks for our dataset: (i) Pragmatics Identification and Reasoning and (ii) Conversational Question Answering. The former task focuses on assessing the capability of models to identify the presence of pragmatic phenomena and their ability to select a suitable answer for such identification. The latter task aims to evaluate the models' adeptness in employing pragmatic reasoning by presenting them with carefully designed questions.

### 4.1   Task 1: Pragmatics Identification and Reasoning (PIR)

In this task, models are provided with dialogues and are required to identify turns whose actual meanings deviate from their literal interpretations, commonly referred to as pragmatic turns. If their selections are accurate, a set of rationales is presented and they are expected to choose the most plausible reason for each pragmatic turn. For each turn, there are 5 candidate reasons available, comprising one gold choice and four disturbing choices. The model's success in this task depends on the precise execution of both steps. We consider three diagnostic settings to test the machine's capability on pragmatic understanding:

- Conversation $\rightarrow$ Pragmatic Turn (**C** $\rightarrow$ **P**). For each instance, models are presented with a dialogue and a specific turn extracted from that dialogue. They are then required to determine whether the given turn qualifies as a pragmatic turn. Consequently, the dataset is flattened to a total of 17,129 instances, with each instance corresponding to a single queried turn. It's important to highlight that turns without pragmatic meanings are also extracted for evaluation.
- Conversation + Pragmatic Turn $\rightarrow$ Rationale (**CP** $\rightarrow$ **R**). In this subtask, we offer the model both the dialogue and the pragmatic turn and it needs to choose the most plausible rationale out of five candidate choices.
- Conversation $\rightarrow$ Pragmatic Turn + Rationale (**C** $\rightarrow$ **PR**) In this subtask, models pre-trained on the previous two subtasks are combined to infer the final results. Specifically, the model obtained from the first subtask is utilized for determining pragmatic turns, while its version finetuned on the second subtask is employed for selecting the most suitable rationale. It is worth noting that, in contrast to **C** $\rightarrow$ **P**, in this subtask, extracted turns are limited to pragmatic turns only.

**Pragmatic Turns and Gold Choices**   Recall that in our data collecting procedure outlined in Sec. 3.1, besides asking workers to select pragmatic turns, they are also instructed to fill in reasons to explain their choices. To simplify the evaluation process, the selected turns are directly utilized as answers for the first subtask, while the reasons provided by the workers serve as the designated correct choices (referred to as "gold choices") for the second subtask.

**Distractor Choice**   As a result of the time-consuming nature of BERTScore [51], an alternative approach is adopted for measuring sentence similarity using Sentence-Transformers [53], which is a significantly faster method. In our methodology, for each gold choice, four alternative choices with high similarity scores are selected from the pool of gold choices to serve as distracting answers. Despite their high similarity scores, upon careful examination within the given context, it's apparent that the distracting answers convey entirely different meanings from the gold answer. This characteristic makes them appropriate components to build our task.

### 4.2   Task 2: Conversational Question Answering (CQA)

The ability to apply pragmatic reasoning is crucial for effective communication and achieving a thorough grasp of the natural language system of human beings. To address this, we propose conversational question-answering, wherein multiple questions are formulated for each dialogue, and an example is shown in Fig. 5. The questions focus on delving deeper into dialogues, often needs insights into intended meanings to answer. ChatGPT plays a pivotal role in question generation and thanks to AMT, we can ensure the collection of high-quality answers. Ultimately, $19,482$ question-answer pairs are assembled.

> **A:** Finally, Ron, lots of talk about Congress releasing the second half of this $700 billion bailout this week. Where do we stand with that?
> **B:** Quite possible that Congress will get that done this week now that Barack Obama has asked George Bush, has asked the current president, to formally put in a request for that money. Congress has got a lot of questions about how this money is going to be spent, as it has questions about how the first half of the money was spent.

> **Rationale:** Barack Obama's request for the $700 billion bailout may expedite the process.

**Q$_1$:** What may expedite the process?
**A$_1$:** Request
**Q$_2$:** Which president-elect requested the $700 billion bailout to be released?
**A$_2$:** Obama

Figure 5: Conversational Question Answering example.

**Question Generation**   ChatGPT is employed to generate questions with prompts consisting of dialogues and human-annotated reasons. We task it to generate questions challenging for individuals who are unaware of the dialogues' intended meanings. More than that, for the convenience of evaluation, the question is also asked to be able to answer within one or two words. Furthermore, to ensure diversity, we instruct ChatGPT to start the questions with "What", "Which" or "How". A preliminary assessment is carried out by sampling a few examples out of the question pool to guarantee quality.

**Answer Collection**   The answers to the questions are obtained through the utilization of AMT. Each worker is provided with a dialogue along with several associated questions and it is requested to answer the questions in one single word. To minimize the potential for misinterpretation, we offer an example coming from our dataset, which is annotated by the author itself. Through the process of human annotation, we consistently evaluate the collected data and reject unqualified answers as well as block workers who fail to meet our standards.

**Statistical features of Questions**   Fig. 3b showcases the diverse range of our questions. These questions encompass a variety of sentence structures, starting with interrogative words: What, Which, and How, and possessing a large diversity of the words that follow the interrogative ones. Apart from the questions, the answer set holds a vocabulary size of 8,179, which is also of great diversity and raises a challenge for models.

## 5   Experiment

### 5.1   Pragmatics Identification and Reasoning

For $\mathbf{C} \rightarrow \mathbf{P}$, we partitioned the dataset into distinct subsets for training, validation, and testing. The training set consists of 13,708 examples, surpassing the 1,361 instances in the validation set and the 2,060 instances in the test set in terms of size. Models are trained on the training set and their

performance is evaluated on the validation set after each epoch to determine the optimal checkpoint. The best checkpoint is subsequently loaded for the final evaluation on the test dataset. The evaluation metric employed is the accuracy score, calculated as the ratio of correct predictions to the total number of instances. Similarly, for the task of **CP → R**, the dataset is also partitioned into training, validation, and test subsets. The respective sizes of these subsets are 5,188, 244, and 1,062 examples. The training and evaluation procedures are identical to those of the previous subtask. For **C →**

Table 3: Pragmatics Identification and Reasoning Results. The numerical results are accuracy scores in their percentage.

| | **C → P** | **CP → R** | **C → PR** |
|---|---|---|---|
| Random | 50 | 20 | 10 |
| BERT$_{base}$ | 63.2 ± 1.1 | 91.3 ± 0.7 | **50.2 ± 6.8** |
| RoBERTa$_{base}$ | 64.4 ± 1.3 | 92.0 ± 0.4 | 50.0 ± 11.28 |
| RoBERTa$_{large}$ | 63.8 ± 0.0 | 60.8 ± 0.5 | 0.0 ± 0.0 |
| GPT-2$_{base}$ | 64.4 ± 0.7 | 90.9 ± 0.9 | 13.06 ± 1.1 |
| DialoGPT$_{medium}$ | 65.0 ± 0.6 | 24.5 ± 1.9 | 3.8 ± 1.5 |
| DeBERTa$_{base}$ | 64.9 ± 0.2 | **92.6 ± 0.6** | 43.9 ± 1.2 |
| ALBERT$_{base}$ | **65.1 ± 0.4** | 90.6 ± 0.2 | 34.9 ± 1.8 |

**PR**, the test sets in previous subtasks are taken for evaluation, and it's worth noting that their test sets consist of exactly the same instances. This design ensures the prevention of any leakage of the test set into the training set, thereby maintaining the integrity of the evaluation process.

**Results and Analysis**   We present four key observations for this task:

◇ The primary performance bottleneck lies in the subtask **C → P**. As shown in Tab. 3, the best model achieves an accuracy score of 92.0% in the **CP → R** subtask, indicating that models possess the capability to reason to some extent. However, when it comes to the **C → PR** task, the best-performing model achieves only 50.2% accuracy, while the highest accuracy achieved in the **C → P** task is 65.0%. The substantial difference between the score of 92.0% and the score of 65.0% suggests that the difficulty in accomplishing the task primarily stems from the **C → P** subtask.

◇ Accumulated variance in the **C → PR** subtask. The models exhibit significant variance in the results of the third subtask, which can be attributed to the variance introduced by its constituent tasks.

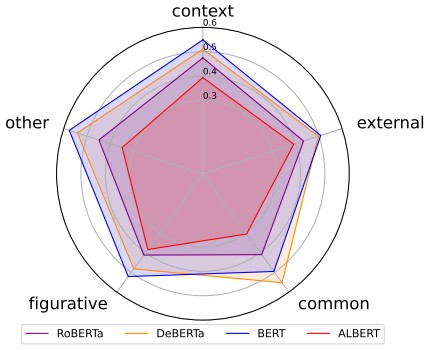

Figure 6: Models performances of different reasoning types. The taxonomy of reasoning types comes from Fig. 2.

◇ The significance of pragmatic awareness in language models. Both the **C → P** and **C → PR** subtasks require pragmatic awareness, and the poor performances of the models on these subtasks highlight their limitations in accurately determining the optimal timing for deploying reasoning abilities.

◇ As depicted in Fig. 6, we assess the performance of existing models across five dimensions by leveraging the taxonomy outlined in Fig. 2. It becomes evident that models exhibit a nearly uniform performance across these dimensions, leading us to the inference that pragmatic reasoning constitutes a cohesive task, and a segregated approach is ill-suited for its treatment.

## 5.2   Conversational Question Answering

Similarly to the previous task, the question-answering dataset is divided into training, validation, and test sets, comprising 15,585, 1,559, and 2,338 instances, respectively. Experimental subjects includes BART [54], T5 [55], UnifiedQA [56], and mT5 [57]. The metric adopted is also accuracy score. Given ChatGPT's impressive performance in MMLU and AI2 Reasoning Challenge [13], we further examine ChatGPT's capability in the context of CQA by prompting it to provide one-word answers to questions. However, due to its uncontrollable nature, the generated answers may not always align with our desired settings. Hence, we introduce two evaluation metrics for ChatGPT: (1) em (exact match), which requires ChatGPT to produce the exact same word as our answer, and (2) pm (partially match), where we consider ChatGPT to be correct as long as our answer appears in its generated output. Two configurations are employed for evaluation. In the first configuration, models receive dialogues and questions while remaining blind to human-annotated rationales. The

Table 4: CQA task results with/without human annotated rationales. The numerical results are accuracy scores in their percentage. em: exact match, pm: partially match.

|  | BART-base | T5-small | mT5-small | UnifiedQA-base | UnifiedQA-large | ChatGPT (em) | ChatGPT (pm) |
|---|---|---|---|---|---|---|---|
| w/o rationale | $20.2 \pm 1.1$ | $24.9 \pm 0.7$ | $19.7 \pm 0.4$ | $28.9 \pm 2.0$ | $32.8 \pm 0$ | 1.0 | **40.6** |
| w/ rationale | $29.6 \pm 0.6$ | $34.1 \pm 1.4$ | $29.8 \pm 0.8$ | $38.8 \pm 0.2$ | $42.4 \pm 0.6$ | 1.5 | **45.1** |
| $\Delta$ | +9.4 (46.5%) | +9.0 (35.8%) | **+10.1 (51.3%)** | +9.9 (34.1%) | +9.6 (29.3%) | +0.5 (60%) | +5 (12.3%) |

second configuration is a contrasting experiment with human-annotated rationales provided. For each configuration, we run each model three times using different random seeds and report the mean and variance of their results as the final outcomes.

**Results and Analysis**   The experimental results are presented in Tab. 4. Our observations can be categorized into three main aspects. Firstly, the importance of pragmatic meaning is proved. As shown in Tab. 4, there exists a notable disparity between the results of models that have access to human-annotated answers and those that do not. On average, the model performances improve by 38.47% after the introduction of human-annotated rationales. Even the lowest-performing model in the initial experiment, mT5-small, demonstrates a 9.4% increase in accuracy. The substantial discrepancy in results between the two configurations underscores the significance of elucidating intended meanings in the development of effective communicators. Second, the models display deficiencies in applying pragmatic reasoning. Our questions are designed to demand a deeper understanding of conversations, however, the models struggle to perform well on our task. The best-performing model, ChatGPT, achieves an accuracy score of 40.6%. It is worth noting that our questions are generated by ChatGPT itself, and our source dataset, Interview, was proposed prior to the emergence of ChatGPT, which means that ChatGPT may have encountered our text during training. These characteristics render its result unsatisfactory. Third, generalization across different types of pragmatic reasoning poses challenges. In this analysis, we focus exclusively on models other than ChatGPT due to the lack of clarity regarding its training process. As demonstrated in  Tab. 4, these models showcase a substantial improvement in performance following the inclusion of human-annotated rationale. The extent of this improvement exhibits slight fluctuations among the various models, suggesting a shared obstacle that hinders their overall performance. Noticed that the models are fine-tuned on a training set that is 5.2 times larger than the test set, we can conclude that achieving effective generalization from one pragmatic reasoning process to another remains a formidable and challenging task.

### 5.3   Zero-Shot Natural Language Inference

We conduct the natural language inference (NLI) task [60] to evaluate the model's comprehension of language and to emphasize the importance of context in pragmatic reasoning. Different from previous tasks, zero-shot NLI task sheds a light on models' initial ability as they are tested without finetuning.  The task involves providing two sentences: a premise and a hypothesis, and models are required to determine the relationship between the two sentences, which can be entailment, contradiction, or neutral. As there are no negative samples in our dataset, we simplify the task by asking the model to judge whether there is entailment. In this task, models are presented with a dialogue, a turn of the dialogue, and an intended meaning, they need to judge whether the turn entails the intended

Table 5: Results of Natural Language Inference Task. $\diamond$: T5-XXL fine-tuned on true NLI mixure [58]. $\dagger$: DeBERTa-v3 trained on MNLI [25] and SNLI [59].

| Method | Acc. |
|---|---|
| Random | 50.0 |
| DeBERTa-Large$^\dagger$ | 44.3 |
| T5-XXL$^\diamond$ | 45.3 |
| ChatGPT | **85.7** |
| ChatGPT w/ CoT | 63.8 |

meaning. Noticed that collected data as described in Sec. 3.1 consists of reasons and implied meanings, to better fit our task, we abandon the reasons and preserve the implied meanings. Models are tested under zero-shot setting, which means that they are not allowed to train on any data before testing. Thus the innate abilities of models play a decisive role. Baseline models include T5 [55], DeBERTa [61], and ChatGPT. It's important to note that ChatGPT and the other two models are tested on different settings. ChatGPT is tested with the whole dialogue and the implied meaning as a prompt. However, to inspect the significance of context, the other two models are only provided with the pragmatic turn and the corresponding pragmatic meaning. ChatGPT is evaluated using two types

of prompts: with and without step-by-step instructions. "`Let's think step by step`" is a prompt discovered [62] to improve the model's reasoning ability; refer to Appendix C for more details.

**Results and Analysis**    Results are listed in Tab. 5. As this task shares similar settings as binary classification, randomized answer accuracy is expected to be 50%. We observe below randomized performance on some previous SOTA models. Note that, each of the data is annotated by two humans, thus it's reasonable to view human performance as 100%. ChatGPT achieves the highest result but still shows a huge gap compared with human annotations. The outcomes highlight the imperfectness of the models' reasoning abilities. For T5-large and DeBERTa, context is blinded, but for ChatGPT, it is reachable. Hence, the performance gap among T5-large, DeBERTa, and ChatGPT shows the importance of context in our task. Interestingly, CoT doesn't offer help to ChatGPT but is harmful to ChatGPT's performance.

# 6    Discussions and Future Work

In this paper, we propose **DiPlomat**, a high-quality manually annotated multi-turn dataset of pragmatic reasoning in conversations. Along with the dataset, we propose two tasks and baselines. Comparing experimental results, we emphasize the nonnegligible impact of contexts and reasoning on building perfect communicators. We also highlight the importance of pragmatic awareness and its bottleneck effect on our tasks. There is still a significant disparity between current performances and established standards.

**Memorization *vs*. Reasoning**    Noticed that models exhibit outstanding performance on $CP \rightarrow R$ of Sec. 5.1. On the contrary, for $C \rightarrow PR$, models achieve poor results. Since the underlying knowledge is consistent for both tasks, the disparity in performance is hypothesized to be attributed to *memorization*. Instead of truly understanding the knowledge, the models tend to memorize patterns.

**Subjectiveness *vs*. Objectiveness**    Ji et al. [63] emphasize the importance of modeling a distribution that encompasses a diverse range of possibilities, rather than solely relying on a single "best" prediction. During the annotation process, we observe a phenomenon that different workers hold diverse opinions regarding pragmatic turns and their intended meanings. Their annotations often exhibit significant variations, sometimes even presenting completely opposing interpretations. We maintain the possibility of subjectiveness with careful task metric design (Sec. 4.1).

**Larger Models *vs*. Smaller Models**    Conventional wisdom suggests that larger models, endowed with a greater number of parameters, generally exhibit superior performance in comparison to their smaller counterparts. However, an intriguing observation comes to the forefront within the context of the Pragmatic Identification and Reasoning (PIR) experiment, as elucidated in Table 3: RoBERTa$_{base}$ surpasses RoBERTa$_{large}$ in performance. We posit that this phenomenon can be attributed to a dual-fold rationale. First and foremost, a divergence emerges between the domain of pragmatic data and the domain of pretraining data for language models. Consequently, larger models exhibit a more substantial and consistent influence of their pretraining data, rendering them less adaptable to domain shifts towards pragmatic data. Secondly, the current landscape of pragmatic data is characterized by its inherent diversity and relative scarcity. This inherent diversity poses a particular challenge for larger models in adapting to such heterogeneity; refer to Appendix A.

**Future Work**    Achieving generalization across multiple types of pragmatic reasoning processes poses significant challenges. Consequently, we propose that the construction of a proficient communicator necessitates the incorporation of methods beyond purely data-driven approaches. Furthermore, the availability of comprehensive evaluation data is of utmost importance. As a result, we target more high-quality datasets and new methods other than data-driven for the problem.

# Acknowledgements

The authors thank Jiaming Yu at Oxford for the preliminary survey, Dr. Xue Feng and Dr. Lifeng Fan at BIGAI for helpful discussions on data collection, Ms. Zhen Chen at BIGAI for designing the teaser figure, Junpeng Li and Dr. Zixia Jia at BIGAI for application experiments. This work presented herein is supported by the National Key R&D Program of China (2022ZD0114900) and the National Natural Science Foundation of China (62376031).

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

# Appendices

## Contents

# A    Hypothetic Explanation for Experimental Results of RoBERTa$_{base}$ outperforms RoBERTa$_{large}$

Conventional wisdom suggests that larger models, with more parameters, tend to outperform smaller models. However, as indicated in Table 3, a notable phenomenon emerges in the context of the PIR experiment: RoBERTa$_{base}$ outperforms RoBERTa$_{large}$. This unexpected observation leads us to formulate a two-fold hypothesis:

1. **domain discrepancy between pragmatic data and LM's pretraining data:** Larger models exhibit a consistent influence of their pretraining phase, whereas smaller models are more susceptible to adaptation during the fine-tuning process. Importantly, beyond the application of semantic knowledge, our pragmatic data introduces deeper challenges for language models, leading smaller models to potentially "superficially" memorize patterns from the fine-tuning data. In contrast, larger LMs tend to adhere more closely to their original pretraining understanding. Similar trends have been observed in other domains, such as multi-step math reasoning [64] and few-shot learning [65].

2. **Low-resource and diversity of pragmatic data:** Pragmatic data is both scarce and diverse. Effectively tackling this data needs language models to move beyond mere semantic pattern memorization and engage in more nuanced reasoning over contextual information, social common sense, and even theory-of-mind reasoning. Such low-resource conditions challenge the conventional fine-tuning requirements for large language models, which typically benefit from substantial data for achieving superior performance.

To empirically substantiate these hypotheses, we undertake a zero-shot version of Task 1 (PIR). The results of this zero-shot experiment are reported in Tab. 6, and when compared to the results presented in Table 3, it becomes evident that the improvement in performance of RoBERTa$_{base}$ following fine-tuning is considerably more pronounced than that of RoBERTa$_{large}$.

Table 6: Zero-Shot PIR

|  | $C \rightarrow P$ | $CP \rightarrow R$ | $C \rightarrow PR$ |
|---|---|---|---|
| RoBERTa$_{large}$ | 61.1 | 21.8 | 1.7 |
| RoBERTa$_{base}$ | 57.2 | 21.5 | 5.1 |

# B    Annotation Details

## B.1    Details For Automatic Selection

Different methodologies are employed to address various pragmatic phenomena. To leverage prior advancements in the field, we begin by segmenting each dialogue into individual utterances. Subsequently, we employ two distinct approaches, namely string matching and pretrained model classification, to identify these phenomena within our source data. In the case of scalar implicature, which exhibits a noticeable pattern characterized by word pairs such as *(some, all)* appearing in adjacent turns of dialogues, we employ string matching to annotate instances of scalar implicature in conversations. Similarly, for popeq implicature, which often features a continuous question mark, we utilize this characteristic as a means of detection. With regards to idioms, which exhibit more evident patterns, we employ the idiom set proposed by Saxena and Paul [18] to conduct searches. For other types of phenomena that lack obvious patterns, we leverage a pretrained RoBERTa base model [49], and fine-tune it for our specific task. The sarcasm dataset by  Misra [66] is used for finetuning the sarcasm model, the MOVER dataset by Zhang and Wan [42] for hyperbole and the ColBERT dataset by Annamoradnejad and Zoghi [20] for paronomasia. Several models have been proposed for metaphor detection, thus we utilize an existing model [67] specifically designed for metaphor identification.

## B.2    Details For Fine-grained Annotation

AMT is integral to our process. To ensure clarity and consistency, we provide explicit instructions to the workers. Additionally, to further elucidate the objectives of our study, we offer illustrative

examples. The task itself is presented below the instructions and examples, with the dialogue and corresponding turn numbers provided for workers to select. Furthermore, as workers check a checkbox, we prompt them to select a confidence score and provide a rationale. In order to strike a balance between our budget, the quality of annotations, and the speed of annotation, we have determined the compensation of $0.1 per completed task. After the annotation process, we collect responses that are assigned with a confidence score of 4 or higher.

Specifically, we surveyed 10 users to accomplish our task. All users can complete a single task within 45 seconds, leading to a wage pay of around 8 dollars per hour, which is about a dollar higher than the federal minimum hourly wage of the United States.

### B.3 Details on Human Refinements

Disturbing choices are chosen based on the BERTScore metric [51]. The rationale with the highest similarity, as determined by other dialogues, is selected and included in the pool of candidate options. The instructions provided to the workers align with those used for Fine-grained Annotation, wherein they are also instructed to assign a confidence score to their responses. The remuneration for workers is set at $0.05 per task.

**AMT Workers Requirements**    In order to guarantee the quality of annotated data, the qualification rules for workers are strict and can be found in Tab. 7.

Table 7: AMT workers requirements

| | |
|---|---|
| **Country**$_{In}$ | United States, Canada, Great Britain, Australia, Singapore, Ireland, New Zealand |
| **# Tasks approved**$_{GreaterThanOrEqualTo}$ | 1300 |
| **Tasks approved Rate**$_{GreaterThanOrEqualTo}$ | 95% |

## C    Experimental Detail

### C.1    Pragmatic Identification and Reasoning (PIR)

**BERT**$_{base}$ [68]    BERT (Bidirectional Encoder Representations from Transformers) is a revolutionary language representation model that has had a significant impact on natural language processing (NLP) tasks. It has achieved remarkable performance across various NLP benchmarks, including question answering, sentiment analysis, named entity recognition, and many others. Its birth brings profound influence on pretrained language models.

**RoBERTa**$_{base}$ **& RoBERTa**$_{large}$ [49]    RoBERTa improves upon BERT by incorporating enhancements such as larger and more diverse training data, longer pretraining duration, dynamic masking, and advanced training strategies. These improvements enable RoBERTa to achieve even better performance on a wide range of NLP benchmarks. While BERT paved the way for contextualized representations in NLP, RoBERTa further refines and pushes the boundaries of language understanding, making it a powerful and preferred choice for many researchers and practitioners in the field.

**ALBERT**$_{base}$ **& ALBERT**$_{large}$ [69]    ALBERT (A Lite BERT) is a highly efficient and compact variant of the BERT model that addresses the computational limitations of the original architecture. It incorporates parameter-reduction techniques to alleviate training time constraints and achieve improved performance compared to BERT.

**DeBERTa**$_{base}$ [61]    DeBERTa (Decoding-enhanced BERT with Disentangled Attention) is a state-of-the-art language representation model that builds upon the BERT architecture and introduces several key innovations, including disentangled attention mechanism. The performance of DeBERTa has been demonstrated to surpass that of BERT on a wide range of NLP tasks.

**GPT2**$_{base}$ [70]    Leveraging transformers decoder, Radford et al. [70] proposed GPT2. It represents a significant breakthrough in natural language processing and generation. One of the most notable features of GPT-2 is its ability to generate coherent and contextually relevant text. Through unsupervised pretraining on a large corpus of internet text, GPT-2 learns to predict the next word in a sequence of text, enabling it to generate human-like responses.

Table 8: Hyperparameters for models on $\mathbf{CP} \rightarrow \mathbf{R}$

| Model | learning rate | batch size | weight decay | epochs |
|---|---|---|---|---|
| BERT$_{base}$ | 5e-5 | 12 | 0.001 | 50 |
| BERT$_{large}$ | 5e-5 | 12 | 0.001 | 50 |
| ALBERT$_{base}$ | 5e-5 | 12 | 0.001 | 50 |
| ALBERT$_{large}$ | 5e-5 | 12 | 0.001 | 50 |
| DeBERTa$_{base}$ | 5e-5 | 12 | 0.001 | 50 |
| RoBERTa$_{base}$ | 5e-5 | 12 | 0.001 | 50 |
| RoBERTa$_{large}$ | 5e-5 | 12 | 0.001 | 50 |
| GPT2$_{base}$ | 0.001 | 8 | 0.01 | 50 |
| DialoGPT$_{medium}$ | 0.001 | 2 | 0.01 | 50 |

Table 9: Batch size for models on $\mathbf{C} \rightarrow \mathbf{P}$

| Model | Batch Size |
|---|---|
| BERT$_{base}$ | 80 |
| ALBERT$_{base}$ | 24 |
| ALBERT$_{large}$ | 24 |
| DeBERTa$_{base}$ | 24 |
| RoBERTa$_{base}$ | 80 |
| RoBERTa$_{large}$ | 24 |
| GPT2$_{base}$ | 24 |
| DialoGPT$_{medium}$ | 8 |

**DialoGPT$_{medium}$ [35]**  DialogGPT is dialogue-oriented GPT. It builds upon the GPT architecture and extends it to support interactive conversations. DialoGPT is trained in a supervised manner using a dialogue dataset, which allows it to understand and generate responses in a conversational context.

The PIR task encompasses three distinct settings: $\mathbf{C} \rightarrow \mathbf{P}$, $\mathbf{CP} \rightarrow \mathbf{R}$, and $\mathbf{C} \rightarrow \mathbf{PR}$. In the $\mathbf{C} \rightarrow \mathbf{P}$ setting, models are trained for 20 epochs, employing a batch size as indicated in Tab. 9, a learning rate of $2e - 5$, and weight decay of $0.01$. As for $\mathbf{CP} \rightarrow \mathbf{R}$ , the hyperparameters adopted are listed in Tab. 8. For the $\mathbf{C} \rightarrow \mathbf{PR}$ setting, there is no training required; instead, we simply load the best checkpoint obtained from the previous training for this task. The concrete implementation is as follows: we initially flatten the test dataset of $\mathbf{C} \rightarrow \mathbf{P}$, ensuring that each instance contains both a dialogue and a pragmatic turn extracted from the same dialogue. As for the test dataset of $\mathbf{CP} \rightarrow \mathbf{R}$, no modifications are made. It should be noted that, following the processing steps, both datasets own the same dialogues and corresponding pragmatic turns, resulting in identical instance numbers. For an instance to be deemed correct, the models must successfully accomplish both component tasks *i.e.* succeed in Identification and Reasoning.

## C.2   Conversational Question Answering (CQA)

**CQA**  ChatGPT was instructed to generate questions for our tasks. The prompt template that starts the questions with "Which" is depicted in Tab. 10. Through this methodology, we collected a total of 19,482 questions. To ensure the reliability of the answers provided to these questions, AMT is utilized. In our experiment, the hyperparameters adopted are illustrated in Tab. 11. To assess the performance of ChatGPT, we conducted testing using the template outlined in Tab. 13.

## C.3   Zero-Shot Natural Language Inference

Details are provided as follows. T5-XXL, and DeBERTa-v3 are tested with the pragmatic turn as premise and implied meaning as a hypothesis. The context is out of reach for these models. In contrast, as shown in Tab. 10, ChatGPT is given the context, and the red line labeled "`Think step by step`" represents two distinct configurations: one with `step-by-step` and one without it.

Table 10: ChatGPT question generation template: using "Which" to start the question.

```
You are sensitive and always view others' words as having some implied
meanings.
For the dialogue between "A" and "B" in this task, we have offered a
statement that is the implied meaning of a turn, please only offer
one reading comprehension question that can be answered with only one
word based on the dialogue and mostly focuses on the turn the statement
mentions.
The question will be tested by only by viewing the dialogue, so please
make the question hard enough that it's impossible to answer without
viewing the statement.
Use "Which" to ask the question!
Following is the dialogue:
{dialogue}
Following is the statement:
{statement}
Use "Which" to ask the question! And please make the question hard
enough that it's impossible to answer without viewing
```

Table 11: Hyperparameters for models on CQA.

| | |
|---|---|
| Training Epoch | 50 |
| Learning Rate | $5.6e-5$ |
| Batch Size | 24 |
| Weight Decay | 0.001 |

Table 12: Test ChatGPT: answer questions with only one word.

```
For the dialogue between "A" and "B" in this task, please answer a
question according to the dialogue with only one word
Following is the dialogue:
{dialogue}
Following is the question: {question}
```

Table 13: ChatGPT test template of Zero-Shot CoT

```
This is a natural language inference task. Given the dialogue context:
{context} Does {pragmatic turn} entails {implied meaning}? Reply
'entails' or 'not entails'.

Think step by step.
```

## D   More Detail on DiPlomat

In this section, we will propose more examples of our dataset in  Tab. 14, Tab. 15, Tab. 16, Tab. 17, and  Tab. 18.

Table 14: Contextual reasoning examples of **DiPlomat**

| | |
|---|---|
| **A**: Yeah. They say that he's the fastest pitcher there ever was. It's just he really couldn't find home plate. I mean, some of the stories you learn about this guy, it reads like fiction. When he was - I think this is around 1960. He's pitching in the minor leagues, and he pitched so fast he ripped the man's ear off.
**B**: Oh.
**A**: Yeah. | **Rationale**: The literal meaning is a simple expression of agreement, while the implied meaning is that the speaker is amazed by the story of Steve Dalkowski's feats. |
| **B**: We're talking about 2. 8 million people. Has the rise of temporary workers figured into, at least, the statistical improvement of the U. S. economy for some people?
**A**: It has. Overall, about one seventh of the total job growth has been in the temp sector. The temp sector is growing nine times faster than the overall private sector as a whole. And the 2. 9 million workers represents a record number, both in the number of temp workers and in the percentage of the economy that they make up.
**B**: You know in "Harvest Of Shame," Edward R. Murrow very famously said, the people we're showing you in this documentary have picked your Thanksgiving bounty with their bare hands, and this is how they live. | **Rationale**: The implied meaning of this turn is to reflect on our reliance on temporary workers in our day-to-day lives. |
| **A**: And so I got up and ran. And it wasn't too far. But I just - at that moment, I thought, I don't want to be shot in the back, and I need to find some cover. And there's really no place to hide. But there are these
**B**: You found a little, like, alcove that you could duck into.
**A**: There was a little alcove, yeah. And I just made myself as small as I could in that little corner. | **Rationale**: The speaker tried to protect itself from danger. |
| **A**: Well, there's a big argument in the United States about this. There's one group of folks who think that engagement policy failed. We engaged with China from 1979 until about 2013 when Xi Jinping came into power. And the idea of engagement was that coevolution was in the American interest as well as in China's interest. And you could bring China along to be a responsible player to some degree.
**A**: Many hardliners in the United States government - and outside and including in the expert community - now claim that engagement was a sucker's game and that we have raised up a tiger which could devour us. But there are different schools of thought about this, and many of us think that we still need to engage with China, albeit more strategically.
**B**: That image of raising a tiger that will devour us is very dramatic. | **Rationale**: The situation is not necessarily an 'either/or' between China and the United States. |

Table 15: Figurative language reasoning examples of **DiPlomat**

| | |
|---|---|
| A: Thank you. How are you?
B: I'm pretty good. Thank you. You must be stuck like glue on this, but, you know, you've played in three World Cups, including one of the wins for the U. S. team in 1999. How would you describe what it's like to be out there on that field in that final game? | **Rationale**: Stuck like glue means to be attached to something, which is a particular issue or a person. |
| **B**: So in terms of what to do about it, we've said Twitter and Facebook have shut down these accounts, which prompts me to wonder - does shutting down a fake account do that much? Can't the Chinese government, if it's determined to go down this path, just open up two new ones in place of the one that was closed?
A: It is a cat-and-mouse game, and the companies are constantly trying to get ahead of it. [···] As you said, they can always set up new accounts. | **Rationale**: Mice are constantly trying to get away from cats and cats are constantly trying to catch mice. In the same way, the Chinese government will always be trying to escape restrictions on social media accounts and media companies will always be trying to find fake accounts. |
| **A**: I really didn't feel safe because the Turkish government is very famous for hunting down those who oppose Erdogan. So, I mean, I just didn't want to really risk my life by going to Europe. But, you know, I talked to my team. I told them all, like, how many times I want to come because I want to be with you guys there, and I want to get a win with you guys. And then, later on, they came back with the news and said, you know what?I think the best decision is if you don't come. Let's just not risk it for one game.
B: Do you feel safe in New York and elsewhere in the U. S. ?
A: I have been getting last two, three days hundreds death threats, but I think I feel safe in America. But anywhere else in the world, I wouldn't really feel safe. | **Rationale**: He is implying that he is still not safe. |

Table 16: Commonsense reasoning examples of **DiPlomat**

| | |
|---|---|
| **B**: Yeah - African-American mayor from Tallahassee.
A: Yes. So this is sort of a test of whether real progressive candidates can win in these sort of purplish states. [...] | **Rationale**: "Purplish" states are not really colored. They refer to US states that are neither clearly Republican (red) nor Democrat (blue) in their voting. |
| **B**: He wrote a lot of letters by hand, didn't he?
A: He wrote tons of letters. I bet there are a hundred thousand - hundreds out there[···] | **Rationale**: tons of letters implies a very large number and not to full a ton. |
| **B**: Well, Pluto's official designation is a dwarf planet. And I have to tell you the people who sent this probe all the way out to Pluto are a little angry about that because when they launched it a decade ago, Pluto was still a planet.
A: (Laughter)
B: It got downgraded in the intervening years.
A: That seems so unfair. | **Rationale**: A is expressing sympathy for the people who sent the probe, showing that they understand why they feel so disappointed. |

Table 17: External knowledge reasoning examples of **DiPlomat**

| | |
|---|---|
| **B**: Inside of his house, family pictures decorate the walls and the fridge. Les has 15 great grandchildren. He grew up in an orphanage, and he couldn't wait to leave to join the military. And so in early 1944, he boarded a ship and crossed the Atlantic Ocean to go to the frontline.
**A**: I loved that sailing on, of course. It was so dramatic. You could see all these ships bobbing up and down on the ocean. And destroyers were weaving in and out of them to make sure they uncovered any mines or anything. | **Rationale**: Sailing across the ocean during wartime was a perilous experience. |
| **A**: . . . equivalent to a nuclear bomb?
**B**: Well, it's about - its equivalent - the energy in that explosion is about 10 times the energy in the first atomic bomb. . . | **Rationale**: The energy released in the explosion is incredibly powerful. |
| **B**: So in your polling, in your research, do you find that it's going to come down to maybe a couple thousand votes from these unaffiliated voters and on what issues?Or will they vote?
**A**: It is likely at the moment to be a very narrow victory. President Bush won in 2004 with five percent. That was 100,000 votes. In other words, if it is one percent, that would be 20,000 votes, and right now, the polls are moving around in just single percentage points. So it could be that narrow.
**B**: Now, I have read that Colorado is going to be this year's Florida and Ohio, that this is going to be the state that decides the election.
**A**: I think it could be, and the interesting thing is that Obama and Palin were both in Jefferson County a couple of days ago, indicating that there may be actually even a county that could be looked at to be beyond an entire state. | **Rationale**: The turn is suggesting that the county of Jefferson in Colorado could be a key factor in deciding the election, despite the fact that it is only one of many counties in the state and there are other swing states in the election. |

Table 18: Others examples of **DiPlomat**

| | |
|---|---|
| **A**: There's that feeling - I mean, so many of us have parents in the industry. I mean, that's what this region is about, especially around Detroit, and Wayne State's in Detroit, the heart of Detroit. So, it's nerve-racking. Everyone is nervous. Everyone doesn't know what's going to happen next. We're all watching the news very closely. But at the same time, it's interesting, because with my generation, we almost seem to, kind of, not be as directly impacted. I mean, our family is, it puts stress on us, but the day to day of the university and the day to day at school doesn't seem to have changed that much.
**B**: I understand you have friends there who are engineering majors. Do they have any sense of what their future looks like, and will be it there in Michigan?
**A**: Everybody is secure in their choices and secure in their decision. Everybody thinks that the industry will come around, especially now with the news that GM is getting money from the government. And everybody is more hopeful, and I mean, the auto industry has always been one of the largest industries and a staple in America, and to think that that industry is just going to vanish, nobody is willing to concede that. | **Rationale**: A believes that the auto industry will not vanish despite the current situation |
| **B**: In the meantime, what more have you learned in your reporting about the death of Carlos Hernandez Vasquez?
**A**: Well, a couple of things. One thing that really stands out is that Carlos Hernandez Vasquez died in a Border Patrol station. The previous migrant children who died were taken to the hospital first; Hernandez Vasquez was not even though immigration authorities clearly knew that he was sick. He was diagnosed with the flu by a nurse practitioner. | **Rationale**: The death of Carlos Hernandez Vasquez could have been prevented if he had been taken to the hospital. |
| **B**: So, how do you and the retired general, James Jones, know each other?
**A**: My gosh, I think - I can't even remember when I first met him. It's been so long ago. I'm sure I met him when he was head of the legislative liaison over the Senate. But I really became acquainted with him when he became a brigadier general, and, of course, I followed his career. Of course, he served very ably as a commandant in the marine corps and then as the European commander, just been with him from time to time. And I just consider him a very good friend. | **Rationale**: A has a high opinion of James Jones' character and career. |

# E    Grice Maxims and Pragmatic Reasoning

The Gricean maxims have garnered substantial attention as a foundational theory within the domain of pragmatics. This theoretical framework comprises four distinct maxims: (1) The Maxim of Quality, (2) The Maxim of Quantity, (3) The Maxim of Relevance, and (4) The Maxim of Manner [16, 15]. In contrast to rigid rules or theorems, the Gricean maxims, which capture the prevalent dynamics of conversations, are susceptible to frequent breaches in the context of human communication. These breaches, stemming from the intricacies of real-world interaction, notably manifest in the violation of one or more of these maxims. Such breaches, aligned with the cooperative principle, give rise to pragmatic phenomena that necessitate the engagement of pragmatic reasoning by recipients of the communication [15].

# F    Computational Resources

For our experiment, we utilized two A100s and one 3090. The majority of our experiments were conducted on the A100s, while for practical reasons, only Unified-QA-base, BART-base, and T5-small were tested on the 3090. It is important to mention that each experiment was run on a single GPU. We record the training time of models in Appendix F.

| Model | $C \rightarrow P$ | $CP \rightarrow R$ | Device |
|---|---|---|---|
| BERT$_{base}$ | 0.8min/epoch | 0.9min/epoch | A100 |
| RoBERTa$_{base}$ | 0.8min/epoch | 0.9min/epoch | A100 |
| RoBERTa$_{large}$ | 2.5min/epoch | 2.8min/epoch | A100 |
| GPT-2$_{base}$ | 5.8min/epoch | 6.2min/epoch | A100 |
| DialoGPT$_{medium}$ | 2.4min/epoch | 4.2min/epoch | A100 |
| DeBERTa$_{base}$ | 0.9min/epoch | 0.9min/epoch | A100 |
| ALBERT$_{base}$ | 0.5min/epoch | 0.8min/epoch | A100 |

Table 19: Training Time of Models

# G    Limitations & Negative Societal Impacts

We acknowledge two limitations in our study: bias and subjectivity. Since our dialogues primarily stem from an interview dataset, a considerable focus is placed on political topics. This is reasonable, as pragmatic phenomena frequently emerge in the statements of politicians to advance their specific goals. However, this focus introduces a certain degree of bias into our dataset. The second limitation relates to the absence of subjectivity. In our methodology, the data undergoes two stages of human annotation, ensuring higher quality and objectivity. However, pragmatic reasoning is inherently subjective, and prioritizing objectivity compromises the preservation of subjectivity, resulting in a limitation in terms of subjectivity coverage. Our dataset exhibits minimal negative societal impacts. This is primarily due to the fact that our dialogues are transcriptions of publicly available TV shows, which inherently limits the potential for negative effects.

# H    Ethics Concern

**Were any ethical review processes conducted (e.g., by an institutional review board)?** Our dataset does not involve human subjects. The dataset has undergone the institute's internal ethical review process.

**Does the dataset contain data that might be considered confidential?** No, the data source derives from an existing public interview dataset.

**Does the dataset contain data that, if viewed directly, might be offensive, insulting, threatening, or might otherwise cause anxiety? If so, please describe why.** Few of the dialogues may talk about offensive topics.

**Does the dataset identify subpopulations (e.g., by age or gender)?** Not explicitly.

**Is it possible to identify individuals (i.e., one or more natural persons) directly or indirectly (i.e., in combination with other data) from the dataset?** Yes, our data contains the names of celebrities.

## I Responsibility & Dataset Liscence

We bear all responsibility in case of violation of rights and our dataset is under the license of CC BY-NC-SA (Attribution-NonCommercial-ShareAlike).

## J Datasheets for Our Dataset

### J.1 Motivation

1. For what purpose was the dataset created? (Was there a specific task in mind? Was there a specific gap that needed to be filled? Please provide a description.)

   This dataset was created to study pragmatic reasoning in dialogues, a specific gap is mentioned above in Appendix G.

2. Who created this dataset (e.g., which team, research group) and on behalf of which entity (e.g., company, institution, organization)?

   This dataset was created by the authors of this paper.

3. Who funded the creation of the dataset? (If there is an associated grant, please provide the name of the grantor and the grant name and number.)

   The institute of the authors funded the creation of the dataset.

4. Any other comments?

   None.

### J.2 Composition

1. What do the instances that comprise the dataset represent (e.g., documents, photos, people, countries)? (Are there multiple types of instances (e.g., movies, users, and ratings; people and interactions between them; nodes and edges)? Please provide a description.)

   An instance of our dataset represent a piece of dialogue. Description is provided in our paper.

2. How many instances are there in total (of each type, if appropriate)?

   Our dataset owns 4,177 dialogues.

3. Does the dataset contain all possible instances or is it a sample (not necessarily random) of instances from a larger set? (If the dataset is a sample, then what is the larger set? Is the sample representative of the larger set (e.g., geographic coverage)? If so, please describe how this representativeness was validated/verified. If it is not representative of the larger set, please describe why not (e.g., to cover a more diverse range of instances, because instances were withheld or unavailable).)

   It is a sample of all possible cases. As pragmatic phenomena aren't proved to be limited, we can't guarantee a full sampling of them.

4. What data does each instance consist of?

   We mention it in our paper.

5. Is there a label or target associated with each instance? If so, please provide a description.

   Yes. The description is in our paper.

6. Is any information missing from individual instances? (If so, please provide a description, explaining why this information is missing (e.g., because it was unavailable). This does not include intentionally removed information, but might include, e.g., redacted text.)

   No. We leverage the original dialogues.

7. Are relationships between individual instances made explicit (e.g., users' movie ratings, social network links)? ( If so, please describe how these relationships are made explicit.)

   No. Instances are weakly related, but focus on the same phenomenon.

8. Are there recommended data splits (e.g., training, development/validation, testing)? (If so, please provide a description of these splits, explaining the rationale behind them.)

Yes. We provide it.

9. Are there any errors, sources of noise, or redundancies in the dataset? (If so, please provide a description.)

Yes. Some workers try to finish the work as quickly as possible, therefore when we ask them to offer a rationale for choosing a certain turn as a pragmatic turn, they simply type an "a" in the box. However, the situation is rare, and we blocked the workers and clean the data out of our dataset.

10. Is the dataset self-contained, or does it link to or otherwise rely on external resources (e.g., websites, tweets, other datasets)? (If it links to or relies on external resources, a) are there guarantees that they will exist, and remain constant, over time; b) are there official archival versions of the complete dataset (i.e., including the external resources as they existed at the time the dataset was created); c) are there any restrictions (e.g., licenses, fees) associated with any of the external resources that might apply to a future user? Please provide descriptions of all external resources and any restrictions associated with them, as well as links or other access points, as appropriate.)

It's self-contained.

11. Does the dataset contain data that might be considered confidential (e.g., data that is protected by legal privilege or by doctor-patient confidentiality, data that includes the content of individuals' non-public communications)? (If so, please provide a description.)

No.

12. Does the dataset contain data that, if viewed directly, might be offensive, insulting, threatening, or might otherwise cause anxiety? (If so, please describe why.)

Yes. Some of the topic are big events, they may be offensive for some people. However, we consider our dataset's offensiveness to be limited, for the source dataset is a TV show transcript.

13. Does the dataset relate to people? (If not, you may skip the remaining questions in this section.)

Yes.

14. Does the dataset identify any subpopulations (e.g., by age, gender)? (If so, please describe how these subpopulations are identified and provide a description of their respective distributions within the dataset.)

No. This is not explicitly identified

15. Is it possible to identify individuals (i.e., one or more natural persons), either directly or indirectly (i.e., in combination with other data) from the dataset? (If so, please describe how.)

Yes; their names are given in running text.

16. Does the dataset contain data that might be considered sensitive in any way (e.g., data that reveals racial or ethnic origins, sexual orientations, religious beliefs, political opinions or union memberships, or locations; financial or health data; biometric or genetic data; forms of government identification, such as social security numbers; criminal history)? (If so, please provide a description.)

Yes. Our dataset may have dialogues talking about religious, politics and so on.

17. Any other comments?

None.

## J.3 Collection Process

1. How was the data associated with each instance acquired? (Was the data directly observable (e.g., raw text, movie ratings), reported by subjects (e.g., survey responses), or indirectly inferred/derived from other data (e.g., part-of-speech tags, model-based guesses for age or language)? If data was reported by subjects or indirectly inferred/derived from other data, was the data validated/verified? If so, please describe how.)

The data all comes from an interview dataset already published. (See our paper)

2. What mechanisms or procedures were used to collect the data (e.g., hardware apparatus or sensor, manual human curation, software program, software API)? (How were these mechanisms or procedures validated?)

   Software program and manual human curation (2 times). See our paper for details.

3. If the dataset is a sample from a larger set, what was the sampling strategy (e.g., deterministic, probabilistic with specific sampling probabilities)?

   Randomly.

4. Who was involved in the data collection process (e.g., students, crowdworkers, contractors) and how were they compensated (e.g., how much were crowdworkers paid)?

   Crowdworkers. They are paid nicely. See Appendix for detail.

5. Over what timeframe was the data collected? (Does this timeframe match the creation timeframe of the data associated with the instances (e.g., recent crawl of old news articles)? If not, please describe the timeframe in which the data associated with the instances was created.)

   The dataset was collected in the early Spring of 2023, which does not necessarily reflect the timeframe of the data collected.

6. Were any ethical review processes conducted (e.g., by an institutional review board)? (If so, please provide a description of these review processes, including the outcomes, as well as a link or other access point to any supporting documentation.)

   No review processes were conducted with respect to the collection and annotation of this data (though review was done for other aspects of this work; see the paper linked at the top of the datasheet).

7. Does the dataset relate to people? (If not, you may skip the remaining questions in this section.)

   Yes.

8. Did you collect the data from the individuals in question directly, or obtain it via third parties or other sources (e.g., websites)?

   Other sources. By curating a published dataset.

9. Were the individuals in question notified about the data collection? (If so, please describe (or show with screenshots or other information) how notice was provided, and provide a link or other access point to, or otherwise reproduce, the exact language of the notification itself.)

   No.

10. Did the individuals in question consent to the collection and use of their data? (If so, please describe (or show with screenshots or other information) how consent was requested and provided, and provide a link or other access point to, or otherwise reproduce, the exact language to which the individuals consented.)

    No. All data are public.

11. If consent was obtained, were the consenting individuals provided with a mechanism to revoke their consent in the future or for certain uses? (If so, please provide a description, as well as a link or other access point to the mechanism (if appropriate).)

    N/A.

12. Has an analysis of the potential impact of the dataset and its use on data subjects (e.g., a data protection impact analysis) been conducted? (If so, please provide a description of this analysis, including the outcomes, as well as a link or other access point to any supporting documentation.)

    No. We consider our dataset having a limited negative effect, for all of our data has been published for more than a year.

13. Any other comments? None.

### J.4 Preprocessing/cleaning/labeling

1. Was any preprocessing/cleaning/labeling of the data done (e.g., discretization or bucketing, tokenization, part-of-speech tagging, SIFT feature extraction, removal of instances, processing of missing values)? (If so, please provide a description. If not, you may skip the remainder of the questions in this section.)

No.

## J.5 Uses

1. Has the dataset been used for any tasks already? (If so, please provide a description.)

   Yes. See our paper for details.

2. Is there a repository that links to any or all papers or systems that use the dataset? (If so, please provide a link or other access point.)

   No.

3. What (other) tasks could the dataset be used for?

   Many more. Such as generation of implied meanings.

4. Is there anything about the composition of the dataset or the way it was collected and preprocessed/cleaned/labeled that might impact future uses? (For example, is there anything that a future user might need to know to avoid uses that could result in unfair treatment of individuals or groups (e.g., stereotyping, quality of service issues) or other undesirable harms (e.g., financial harms, legal risks) If so, please provide a description. Is there anything a future user could do to mitigate these undesirable harms?)

   No.

5. Are there tasks for which the dataset should not be used? (If so, please provide a description.)

   No.

6. Any other comments?

   None.

## J.6 Distribution

1. Will the dataset be distributed to third parties outside of the entity (e.g., company, institution, organization) on behalf of which the dataset was created? (If so, please provide a description.)

   Yes, the dataset is freely available.

2. How will the dataset will be distributed (e.g., tarball on website, API, GitHub)? (Does the dataset have a digital object identifier (DOI)?)

   On our website.

3. When will the dataset be distributed?

   It's already been distributed.

4. Will the dataset be distributed under a copyright or other intellectual property (IP) license, and/or under applicable terms of use (ToU)? (If so, please describe this license and/or ToU, and provide a link or other access point to, or otherwise reproduce, any relevant licensing terms or ToU, as well as any fees associated with these restrictions.)

   The dataset is licensed under a CC license.

5. Have any third parties imposed IP-based or other restrictions on the data associated with the instances? (If so, please describe these restrictions, and provide a link or other access point to, or otherwise reproduce, any relevant licensing terms, as well as any fees associated with these restrictions.)

   Not to our knowledge.

6. Do any export controls or other regulatory restrictions apply to the dataset or to individual instances? (If so, please describe these restrictions, and provide a link or other access point to, or otherwise reproduce, any supporting documentation.)

   Not to our knowledge.

7. Any other comments?

   None.

### J.7 Maintenance

1. Who is supporting/hosting/maintaining the dataset?

   The authors.

2. How can the owner/curator/manager of the dataset be contacted (e.g., email address)?

   We will post our email address.

3. Is there an erratum? (If so, please provide a link or other access point.)

   Currently, no. As errors are encountered, future versions of the dataset may be released (but will be versioned). They will all be provided in the same location.

4. Will the dataset be updated (e.g., to correct labeling errors, add new instances, delete instances')? (If so, please describe how often, by whom, and how updates will be communicated to users (e.g., mailing list, GitHub)?)

   Yes.However, the frequency isn't determined, and we'll publish the updated dataset on the same website if an renewal occurs, and we'll anounce it on the website.

5. If the dataset relates to people, are there applicable limits on the retention of the data associated with the instances (e.g., were individuals in question told that their data would be retained for a fixed period of time and then deleted)? (If so, please describe these limits and explain how they will be enforced.)

   No.

6. Will older versions of the dataset continue to be supported/hosted/maintained? (If so, please describe how. If not, please describe how its obsolescence will be communicated to users.)

   Yes. The older versions of the dataset will be available on the website.

7. If others want to extend/augment/build on/contribute to the dataset, is there a mechanism for them to do so? (If so, please provide a description. Will these contributions be validated/verified? If so, please describe how. If not, why not? Is there a process for communicating/distributing these contributions to other users? If so, please provide a description.)

   Yes. They can email us.

8. Any other comments?

   None.

