# OpenReview forum: "Diplomat: A Dialogue Dataset for Situated PragMATic Reasoning"
_NeurIPS.cc/2023/Track/Datasets_and_Benchmarks — NeurIPS 2023 Datasets and Benchmarks Poster_

### Official Review · Reviewer_hCbW · 2023-07-19
**The comprehensive dataset and challenging tasks that have significant potential for advancing pragmatic reasoning in conversational agents.**

**Rating:** 6
**Confidence:** 4

**Strengths:**

The dataset, PIR task, and CQA task each possess significant merits.

The authors' meticulous effort in dataset creation is commendable, as evidenced by the extensive review of prior works and the rigorous curation process. I was able to download and inspect the dataset, and the authors provided the detailed explanations in the Appendix.

The PIR task has informative subtasks, which were able to highlight the limitations of current models in understanding pragmatic awareness. This task offers valuable insights that can guide future research in the field of pragmatic reasoning.

The CQA task presents an intriguing challenge for agents, and even for non-native English speakers like myself. It offers promising research opportunities to delve into detailed dialogues and process implied meanings with language models.


**Additional Feedback:**

Typos
* Page 6, line 198: unnecessary “and” between “required to determine” and “whether”
* Page 9, line 331: redundant comma before “In this task,”

Many references are incomplete: for example, 10, 11, 13, 18, 22, 34, 35, 36, 37. Please check.


**Clarity:**

The paper is overall well-written, but it could be further improved with some minor polishing.


**Correctness:**

The dataset was created rigorously through automatic selection, fine-grained annotation, and human refinement. More details on the human refinement process would be helpful.

The evaluation tasks, PIR and CQA, are well-designed and provide convincing results. Connecting the NLI task to pragmatic reasoning would be helpful.


**Documentation:**

The dataset is freely available and easy to access. The authors have provided the code for running experiments, which I hope will be made publicly available soon with helpful documentation.


**Ethics:**

The authors provided compensation of $0.1 per completed task. To determine if this amount aligns with the minimum wage, additional information is required, such as the median total compensation of each worker, estimated/median time spent on tasks, and the percentage of rejected work and blocked workers.

The authors stated that they did not conduct any ethical review processes. However, according to the NeurIPS Code of Ethics, data annotation and refinement by crowdworkers may be categorized as **research "involving" human participants**. The authors should consult their institutions and follow their guidance.

The below is the excerpt from the NeurIPS Code of Ethics (https://neurips.cc/public/EthicsGuidelines)

***Research involving human participants**: if the research presented involves direct interactions between the researchers and human participants or between a technical system and human participants, authors are required to follow existing protocols in their institutions (e.g. human subject research accreditation, IRB) and go through the relevant process.*


**Limitations:**

I have specific suggestions and questions.

1. Clarify the dataset name.
2. Explicitly state how the NLI task proposed in this study contributes to pragmatic reasoning and differs from previous research.
3. In Tables 3 and 5, the chance-level performance is presented as “Random”. Are the number of cases balanced?
4. I could not follow the details of the Human Refinement section. Why are their multiple gold choices? Is it because two different workers provided reasons for the same dialogue, and if there was a careless worker providing the reason, the refinement accuracy would be lower?
5. When referring to the Supplementary material, clearly identify the relevant sections or tables to facilitate easy access and comprehension for readers.

---

This suggestion would be out of scope for this paper and perhaps more suitable for social science venues. However, it’d be fascinating to see human performance on the PIR and CQA tasks, particularly from non-native English speakers who may encounter challenges due to different contexts and cultural backgrounds.


**Opportunities For Improvement:**

I have few suggestions that I believe may strengthen the paper.

First, the paper would benefit from further polishing. For example, there are some inconsistencies such as using both "Diplomat" and "PunchLine" as the dataset's name, which should be clarified. The connection between the natural language inference (NLI) task and the rest of the paper is not clearly established. Also, there are some missing details in the description of the dataset construction and human refinement sections.

Second, exploring the performance of recent larger open-source models on the proposed tasks would be significant. Assessing how these models handle pragmatic reasoning and identifying any specific features that contribute to improved performance would yield valuable insights.

Third, the authors categorized the dataset into five reasoning types -- contextual reasoning, figurative language understanding, commonsense reasoning, external knowledge reasoning, and others. Further analysis and evaluation of models' performance on each reasoning type may lead to interesting future developments. This opportunity can be readily explored as the dataset has already been categorized.

Although the authors have stated the limitation, I’d like to reiterate that understanding pragmatics in this dataset is inherently challenging due to the subjective nature of pragmatics. Moreover, certain reasoning types such as "Commonsense" and "External Knowledge" may require knowledge specific to the cultural context of educated individuals in American society. It would be beneficial to make such knowledge explicit in the dataset, so that future research can take advantage of such extra information.


**Relation To Prior Work:**

This study provides a comprehensive review of prior works and datasets on pragmatic understanding. The review led to the development of an impressive dataset and tasks, which have significant potential for advancing pragmatic reasoning in conversational agents.


**Summary And Contributions:**

The paper introduces Diplomat, a comprehensive annotated dataset derived from the NPR interview dataset. It also presents two challenging tasks: pragmatic identification and reasoning (PIR) and conversational question answering (CQA) based on the dataset.

The PIR task focuses on identifying pragmatic turns in a dialogue, and it includes multiple subtasks: conversation -> pragmatic turn (C -> P), conversation + pragmatic turn -> rationale (CP -> R), and conversation -> pragmatic turn + rationale (C -> PR). The authors found that the current language models struggle with the C -> P subtask, indicating a lack of pragmatic awareness in these models.

The CQA task requires models to answer questions about a dialogue without access to its intended meaning. The authors found that the current language models also did not perform well on this task, highlighting the challenges of applying pragmatic reasoning in conversational question answering.

Overall, this study presents an impressive combination of a comprehensive dataset, thoughtfully-designed, challenging tasks, and solid experiments on pragmatic reasoning, which has significant potential for advancing pragmatic reasoning in conversational agents.

---

> ### Author Response · Authors · 2023-08-24
> **Response to Reviewer hCbW (1)**
>
> Thank you for your carefull review. We have provided answers for the questions, we hope that the answers are satisfying. Please let us know if there are still problems, we are delighted to have deeper and interesting discussion! Many thanks!
> > First, the paper would benefit from further polishing. For example, there are some inconsistencies such as using both "Diplomat" and "PunchLine" as the dataset's name, which should be clarified. The connection between the natural language inference (NLI) task and the rest of the paper is not clearly established. Also, there are some missing details in the description of the dataset construction and human refinement sections.
>
> Thanks for you suggestion! We've tried our best to polish our paper, and the result is demonstrated in the revised submission. We've corrected the typos and the correct name for our dataset is "Diplomat". The relation of NLI task and the rest of the paper is clarified in the revised submission. We've added more details for the dataset construction and human refinement section, and we hope it's clear enough. Many thanks!
>
> > Second, exploring the performance of recent larger open-source models on the proposed tasks would be significant. Assessing how these models handle pragmatic reasoning and identifying any specific features that contribute to improved performance would yield valuable insights.
>
>
> Thanks for pointing out! Pragmatic Awareness is important. We observe that current highest challenge lies in Pragmatic Awareness of LLMs. These models can't explicitly reason and aware of the pragmatic turn occured in the history. Enhancing models' ability of awareness can largely improve performance (As seen in C->PR of Table 3).
>
>
>
>
> > Third, the authors categorized the dataset into five reasoning types -- contextual reasoning, figurative language understanding, commonsense reasoning, external knowledge reasoning, and others. Further analysis and evaluation of models' performance on each reasoning type may lead to interesting future developments. This opportunity can be readily explored as the dataset has already been categorized.
>
>
> Thanks so much for the suggestion! We conduct experiments of Task1: C->PR and report the results in the **Figure 6** of the **revised submmision**.
> - It's demonstrated that models' performances are unsatisfying in each dimension.
> - DeBERTa model which performs the best in Commonsense type was beaten by other models in the other types, indicating that **generalization among different reasoning types is uneasy**.
>
>
>
>
>
> > Although the authors have stated the limitation, I’d like to reiterate that understanding pragmatics in this dataset is inherently challenging due to the subjective nature of pragmatics. Moreover, certain reasoning types such as "Commonsense" and "External Knowledge" may require knowledge specific to the cultural context of educated individuals in American society. It would be beneficial to make such knowledge explicit in the dataset, so that future research can take advantage of such extra information.
> > Clarify the dataset name.
>
> **A**: Diplomat. There are two aspects considering using this name:
>
> 1. We choose Diplomat as an acronym of "**Di**a**lo**gue Dataset for Situated **P**rag**mat**ic Reasoning"
> 2. The figure of a diplomat often corresponds frequently using pragmatic sentences. As is known that a diplomat tends to hide his/her true intention under the words, which accords with the characteristics of our dataset
>
> Thanks for point out! We've added this part to the revised submission, and we'll correct the typos!

---

> > ### Author Response · Authors · 2023-08-24
> > **Response to Reviewer hCbW (2)**
> >
> > > Explicitly state how the NLI task proposed in this study contributes to pragmatic reasoning and differs from previous research.
> >
> > **A**: Thanks for asking! The NLI task serves as a direct zero-shot testing method of models' pragmatic reasoning ability. Following is a clear clarification of the task:
> > 1. **Zero-Shot NLI Task of Ours**: Generally, the NLI task can be described as:" It(NLI task) involves providing two sentences: a premise and a hypothesis, and models are required to determine the relationship between the two sentences, which can be entailment, contradiction, or neutral."(*line 328*) In our setting, we simply ask the model to judge whether the relationship is **entailment** or not.
> > 2. **Relationship between Zero-Shot NLI task and Pragmatic Reasoning**: Noticed that we select the rationales that are suitable for being a hypothesis, and the model is asked to check whether it's possible to derive the selected rationales from their original sentences. To know that the derivation is possible, the ability to do pragmatic reasoning is essential.
> > 3. **Difference between Zero-Shot NLI task and C->PR**: C->PR is a better task for Pragmatic Reasoning than Zero-Shot NLI task, as it check models' ability more comprehensively. However, apart from testing models' ability and providing evidence for our statement, the results of Zero-Shot NLI task also support the validity of our dataset annotation.
> >
> >
> >
> > > In Tables 3 and 5, the chance-level performance is presented as “Random”. Are the number of cases balanced?
> >
> > **A**: The values of "Random" are theoritical results rather than empirical ones.
> >
> > - Table 5 demonstrates the result of **zero-shot** NLI. In our dataset, we don't have negative examples for NLI, therefore for a random model, it will choose "entail" or "not entail" each with 50%, therefore the expected accuracy of the task is 50%.
> > - For Table3, the subtasks are multiple choice task, and we randomly shuffle the order of choices for balancing. The calculation method of "Random" in table 3 is similar to that of table 5.
> > > Ethics:
> > >The authors provided compensation of $0.1 per completed task. To determine if this amount aligns with the minimum wage, additional information is required, such as the median total compensation of each worker, estimated/median time spent on tasks, and the percentage of rejected work and blocked workers.
> >
> >
> > Thanks so much for pointing out! We surveyed 10 users to accomplish our task ahead of Turker experiments. All users can complete a single task within 45 seconds, leading to a wage pay of around 8 dollars per hour, which is about a dollar higher than the federal minimum hourly wage of the United States.  Our payment matches the working payload of annotation tasks. We have surveyed Amazon Turk market price of similar tasks (such as reading comprehension, question answering, and language generation...), the average pay for each assignment is around 0.08 dollars, thus we choose 0.1 dollars per assignmentand strict qualification requirements (Refer to Table 7 of Appendix B) to ensure the quality of annotation.
> >
> >
> >
> > >The authors stated that they did not conduct any ethical review processes. However, according to the NeurIPS Code of Ethics, data annotation and refinement by crowdworkers may be categorized as research "involving" human participants. The authors should consult their institutions and follow their guidance.
> >
> > >The below is the excerpt from the NeurIPS Code of Ethics (https://neurips.cc/public/EthicsGuidelines)
> >
> > >Research involving human participants: if the research presented involves direct interactions between the researchers and human participants or between a technical system and human participants, authors are required to follow existing protocols in their institutions (e.g. human subject research accreditation, IRB) and go through the relevant process.
> >
> > Thanks so much for pointing out! We carefully review the NeurIPS Code of Ethics, we consider our data collection doesn't involve **direct interactions** between the researchers and human participants or between a **technical system** and human participants. We've carefully reviewed our annotations to avoid toxic data. We only leverage Amazon Turk for data annotation, we strictly follow the rules of Amazon Turk (https://www.mturk.com/worker/acceptable-use-policy) to publish assignments.

---

> > > ### Author Response · Authors · 2023-08-24
> > > **Response to Reviewer hCbW (3)**
> > >
> > > > I could not follow the details of the Human Refinement section. Why are their multiple gold choices? Is it because two different workers provided reasons for the same dialogue, and if there was a careless worker providing the reason, the refinement accuracy would be lower?
> > >
> > > **A**: The reason for having multiple gold choices is because: in step 2 (Fine-grained Annotation), we ask 2 workers to offer their answers, and step3 is more like a double-check. (1) If there was a careless worker in step 2, a double check in step 3 (Human-Refinement) may improve the accuracy of our dataset. (2) If there was a careless worker in step 3, the answers coming from step 2 may be sufficient. (3) If there was a careless worker in step 2 and another careless one in step 3, we still have a second worker in step 2. (4) If three of them are careless, we believe that this situation is rare, as the author itself checks the answers of step 2,3 from time to time and blocks careless workers. We hope we have clarified the problem, many thanks for pointing it out!
> > >
> > > > When referring to the Supplementary material, clearly identify the relevant sections or tables to facilitate easy access and comprehension for readers.
> > >
> > >
> > > **A**:
> > > Thanks so much for the advice, we've improved our writing of Supplementary Materials!
> > >
> > > > Typos
> > >
> > > > Page 6, line 198: unnecessary “and” between “required to determine” and “whether”
> > > > Page 9, line 331: redundant comma before “In this > task,”
> > >
> > > Thanks for reminding us! We've corrected these typos in our revised submission.
> > >
> > > > Many references are incomplete: for example, 10, 11, 13, 18, 22, 34, 35, 36, 37. Please check.
> > >
> > > Thanks for pointing out! We've corrected them!

---

### Official Review · Reviewer_tasC · 2023-07-20
**Promising Advances in Pragmatic Reasoning**

**Rating:** 7
**Confidence:** 4
**Correctness:** Yes.
**Clarity:** The paper is well-written and easy to…

**Strengths:**

- **Relevant and Underexplored Problem**: The paper addresses the problem of pragmatic reasoning, which is of significant interest to the NLP community. It sheds light on an underexplored area, highlighting its importance for advancing natural language understanding.
- **Rich Diversity and Real-life Coverage**: The Diplomat dataset demonstrates its robustness through rigorous filtering, showcasing a wide array of topics and rich diversity sourced from real-life corpora. This comprehensive coverage enhances the dataset's applicability and relevance to real-world scenarios.
- **Emphasizing Context Understanding**: The experimental results presented in the paper emphasize the critical importance of context understanding. They highlight the limitations of current powerful language models, such as ChatGPT, in this aspect, paving the way for further advancements.
- **Clarity and readability**: The paper is thoughtfully structured and presented in a manner that makes it mostly easy to follow. Its clear organization and coherence facilitate a smoother understanding of the research and its implications.

**Additional Feedback:**

No

**Documentation:**

Yes.

**Limitations:**

Same as above.

**Opportunities For Improvement:**

To enhance the work, it would be valuable to include a dedicated section on "Safety and Ethical Implications" in the paper. This section can assess the potential ethical concerns and safety considerations associated with the Diplomat dataset. One approach to further filtering unsafe responses could involve utilizing a toxicity classifier. By incorporating such measures, the research can address critical issues related to responsible AI development and ensure the dataset's suitability for various applications.






**Relation To Prior Work:**

Yes.

**Summary And Contributions:**

The paper introduces Diplomat, a novel benchmark that combines pragmatic reasoning and situated conversational understanding, drawing insights from cognitive linguistics and conversational modeling. The dataset is formed through three curation steps: Automatic Selection, Fine-grained Annotation, and Human Refinement, resulting in 4K multi-turn dialogues extracted from a real-world interview dataset spanning two decades of radio programs.

The Automatic Selection process eliminates redundancy in the data, while Fine-grained Annotation employs Amazon Mechanical Turk (AMT) for detailed annotation of pragmatic turns, ensuring high-quality selections. This leads to ~6K dialogues. In the Human Refinement step, poorly annotated dialogues are further filtered out to improve data quality.

Diplomat presents two tasks: Pragmatic Identification and Reasoning, focusing on the model's ability to identify pragmatic phenomena and select appropriate answers, and Conversational Question Answering (CQA), comprising ~19K questions with human-written answers derived from the collected dialogues. The study reveals that state-of-the-art language models exhibit poor performance when dealing with this nuanced and subjective topic. Additionally, the paper emphasizes the importance of context understanding in fostering benign human-machine interactions.

---

> ### Author Response · Authors · 2023-08-24
> **Response to Reviewer tasC**
>
> Thanks for acknowledging and appreciating our work! We hope our dataset can make a step forward from current language models to  general emotional intelligence models. We are delighted to know that you are interesting in our work, and we are more than happy to have further discussions!
> > To enhance the work, it would be valuable to include a dedicated section on "Safety and Ethical Implications" in the paper. This section can assess the potential ethical concerns and safety considerations associated with the Diplomat dataset. One approach to further filtering unsafe responses could involve utilizing a toxicity classifier. By incorporating such measures, the research can address critical issues related to responsible AI development and ensure the dataset's suitability for various applications.
>
>
> **A**: We have carefully list "Ethical Concerns" in Section G of Supplementary Material. We intentionally choose a publicly released dataset to avoid further ethical problems. We carefully reviewed our annotations to ensure that there is no toxic annotations. Thank you for the suggestion, and we'll adopt the automatic method.

---

### Official Review · Reviewer_r1EM · 2023-07-21
**Unclear contribution**

**Rating:** 5
**Confidence:** 4

**Strengths:**

I liked the focus on dialogues from real-world interviews.

**Additional Feedback:**

- Why not ask models to generate rationales in CP->R? It should be within reach of recent large language models.
- Why not ask models to identify when pragmatic reasoning is needed in C->R? it would be more realistic

**Clarity:**

The paper, is generally clear. However, I have a couple of questions on the Table 1:
- what is the column manually referring to?
- what is the difference between "diverse" and "open" in terms of domain?

Also in section 3.1, the "disturbing" choices would be better called "distractor" choices.

On line 307, which model performances improve by 38.47%?

**Correctness:**

- on line 133 it is stated that the reasons are required to be no more than 8 words, however, the rationales in figure 2 seem to be much longer, e.g. the last one has 17 words. Which is correct? Enforcing a word limit so low sounds rather restrictive, how was it determined?

- given the difficulty and the subjectivity of the task, I was expecting to see some inter-annotator agreement study to verify that the guidelines were interpreted correctly, however I couldn't find it.

- in the human refinement process, if the subjects choose the distractor/disturbing choice their work is rejected. But couldn't it be the case that the distractor/disturbing choice just happened to be equally good as the original, given the open-ended nature of the task?

**Documentation:**

The dataset is adequately described.

**Ethics:**

I can't see any issue since the data used is already part of a different dataset that is publicly available and the extra annotation doesn't involve personal information.

**Limitations:**

I couldn't see anything critical missing.

**Opportunities For Improvement:**

- A key issue for me is the justification for the reasoning types considered. On line 38 it is stated that these choices are motivated by theories of cognitive linguistics and conversational modelling, however no citations are offered. Thus it is not clear whether the core components of real-life conversation pragmatic reasoning are covered. Furthermore, some previous work is cited on the figurative language, but this for me brings the question: why should metaphors, idioms and puns be treated as one category, and not separately as these works suggest? On the whole, I was expecting Grice's maxims to be relevant to this, but they were not mentioned at all.

**Relation To Prior Work:**

- It is unclear what was missing from previous work which calls for this new dataset. Annotated data is always welcome, but what does this enable us to do that was not previously possible? Literal interpretations of figurative language and conversational question answering for example have their own, separate datasets. What is the benefit of having them on one?

- Previous work, e.g. GRICE and Mutual are critized for their lack of diversity, but then we see that the proposed dataset is about politics and world events, so perhaps the diversity of each dataset should be measured and compared for a fairer assessment.

- For task 2 I have some concerns on both the question generation and the answers. Why is it useful/interesting/etc. to have systems developed to answers questions that were generated by ChatGPT, as opposed to humans themselves? How do we know these questions are useful? And how did you ensure the quality of the answers by AMT? Quality issues are known, but they have now been exacerbated as crowdworkers now use chatGPT for their work; see here: https://arxiv.org/abs/2306.07899. Again inter-annotator agreement would have been appropriate

- Also how did you assess the challenge posed by the questions? You need to compare against different datasets.

**Summary And Contributions:**

This paper proposes a dialogue dataset for situated pragmatic reasoning. It is a subset of a previous dataset containing interviews from NPR radio, which have been annotated for two tasks: pragmatic identification and reasoning, and conversational question answering. The former is a classification task with textual explanations, and the latter consists of questions and answers on the conversations. Models are proposed and evaluated for the two tasks, using established practices.

Following the responses of the authors, I decided to keep my score. The dataset proposed is OK, but not good enough for a top venue such as NeurIPS, since the quality checks were not conducted and the evaluation is not on human pragmatic reasoning.

---

> ### Author Response · Authors · 2023-08-24
> **Response to Reviewer r1EM (1)**
>
> Thank you for the careful review and insightful comments! We discuss the problems carefully and through our answers we clarify some misunderstandings. We hope our point-by-point responses have addressed all of your concerns. We are more than willing to involve in discussions if you have any further questions.
>
> **Opportunities For Improvement:**
>
> > A key issue for me is the justification for the reasoning types considered. On line 38 it is stated that these choices are motivated by theories of cognitive linguistics and conversational modelling, however no citations are offered. Thus it is not clear whether the core components of real-life conversation pragmatic reasoning are covered.
>
> **A:** Thanks so much for your suggestions!
> We have mentioned a few cognitive linguistics and conversational modeling references [2,3,4] in our paper, and we've added them to line 38 in the revision.
>
> [2] Alan P Fiske. The four elementary forms of sociality: framework for a unified theory of social relations, 1992.
>
> [3] John A Bargh and Tanya L Chartrand. The unbearable automaticity of being. American psychologist, 1999.
>
> [4] Edward Finegan. Language: Its structure and use. Cengage Learning, 2014
>
>
> More classical references on "cognitive linguistics and conversational modeling" are listed as follows:
>
>
> 1. **Cognitive Linguistics**:
> - Daniel l. Schacter, Daniel t. Gilbert, Daniel M. WeGner, MattheW K. NocK. Psychology, 2016
> - Richard J.Gerrig. Psychology and Life, 1975
> -  Cognitive Linguistics: The Experiential Dynamics of Metaphor. F. ELIZABETH HART, 1995
> 3. **Conversational Modeling**:
> - Keith Brown, Eve V.Clark, Jim Miller, Lesley Milroy, Geoffrey K. Pullum, and Peter Roach. Meaning in Language: An Introduction to Semantics and Pragmatics, 2004
> - Herbert H. Clark and Bridget Bly. Pragmatics and Discourse, 1995
> - Michael C. Frank and Noah D. Goodman. Predicting Pragmatic Reasoning in Language Games, 2016
>
> We would like to explore more related work and please let us know if we missed any references.
>
>
> > Furthermore, some previous work is cited on the figurative language, but this for me brings the question: why should metaphors, idioms and puns be treated as one category, and not separately as these works suggest?
>
> **A:** Thanks for asking the question, and I'd like to answer the question from 3 aspects:
> 1. **Cognitive Perspective:** Intuitively, human beings handle these together: through the process of language acquisition, human beings gain these abilities without clear order. Such phenomena can also be revealed in the designation of classical pragmatic literature [a].
> 2. **Modeling Perspective:** All of these can be modeled as a process to shift from implicit to explicit, which is aligned with current unfied training paradigm of large language models. By modeling these individual aspects together, we aim to persue a more general language model on pragmatic reasoning.
>
> 3. **Boundary Perspective:** The boundary among figurative tasks is not clear, such as "The classroom was a zoo." can be a metaphorical expression and can also be a pun under the circumstance of teaching noisy animals in the zoo. Apart from that, pragmatic reasoning is a complex phenomenon, we find out that some of the reasoning processes can't be sorted out according to the taxonomy of Pragmatics.
>
> By combining all these aspects together, we hope to provide a window for enhancing the generality of language models.
>
> [a] *Keith Brown, Eve V.Clark, Jim Miller, Lesley Milroy, Geoffrey K. Pullum, and Peter Roach*, Meaning in Language: An Introduction to Semantics and Pragmatics

---

> > ### Author Response · Authors · 2023-08-24
> > **Response to Reviewer r1EM (2)**
> >
> > > I was expecting Grice's maxims to be relevant to this, but they were not mentioned at all.
> >
> >
> > **A:**  Thanks so much for pointing it out! We summarize the relationships between Grice's maxims and Pragmatic Reasoning:
> > 1. **Grice's Maxims:** (1) Maxim of quality (2) Maxim of quantity (3) Maxim of relevance (4) Maxim of manner
> > 2. **Relationship of Grice's Maxims and Our work:** It's clear that Grice's Maxims aren't rules for human communication. On the contrary, it's often violated. In the case of PopeQ, the maxim of relevance can be violated, such as the dialogue given below:
> > > Mom: Are you sure you can take care of yourself this weekend?
> > > Bob: Mom, can a duck swim?
> >
> > Under the Principle of Cooperation, when a maxim is violated, a pragmatic phenomenon occurs [a][b].
> >
> > We have added the above discussion to our revised paper. (Section 2: Related Work): "`The Gricean maxims have garnered substantial attention as a foundational theory within the domain of pragmatics. This theoretical framework comprises four distinct maxims: (1) The Maxim of Quality, (2) The Maxim of Quantity, (3) The Maxim of Relevance, and (4) The Maxim of Manner[a][b]. In contrast to rigid rules or theorems, the Gricean maxims, which capture the prevalent dynamics of conversations, are susceptible to frequent breaches in the context of human communication. These breaches, stemming from the intricacies of real-world interaction, notably manifest in the violation of one or more of these maxims. Such breaches, aligned with the cooperative principle, give rise to pragmatic phenomena that necessitate the engagement of pragmatic reasoning by recipients of the communication.[a][b]`"
> >
> >
> > [a] Analyzing Meaning - An Introduction to Semantics and Pragmatics---by *Paul Kroeger*
> >
> > [b]Meaning in Language: An Introduction to Semantics and Pragmatics---by *Keith Brown, Eve V.Clark, Jim Miller, Lesley Milroy, Geoffrey K. Pullum, and Peter Roach*
> >
> > **Correctness:**
> >
> > > on line 133 it is stated that the reasons are required to be no more than 8 words, however, the rationales in figure 2 seem to be much longer, e.g. the last one has 17 words. Which is correct? Enforcing a word limit so low sounds rather restrictive, how was it determined?
> >
> > **A**: Thanks for pointing this out! There is no such maximum amount limit during our curation. This is our initial setting and it's set for our preliminary test, but later we remove the constraint, and the setting is not leveraged during formal curation making our dataset better at testing models' ability. We've corrected this typo in the revised submission. Thanks!
> >
> > > given the difficulty and the subjectivity of the task, I was expecting to see some inter-annotator agreement study to verify that the guidelines were interpreted correctly, however I couldn't find it.
> >
> >
> > **A:** In step 2 (Fine-grained Annotation), for each data task, **two** workers are included in the annotation process. While in step 3, a **third** worker is asked to judge the answers of previous workers. The collected human-annotated reasons of step 2 are transformed into choices, utilizing a template format: `[turn {turn_id}: {reason}]`, and the worker of step 3 is ordered to check the choices. Apart from that, to avoid workers choosing without careful reading, we add disturbing choices into the choice pool. It's worth noting that the third worker is not allowed to offer a new idea of the pragmatic turns, he/she can only agree or disagree with previous works. Therefore, under this setting, the annotations of the third worker represent the agreement between he/she and the first two workers. Furthermore, we report in line 153: "`Through this refinement process, 1,692 dialogues are filtered out, while 4,177 dialogues are preserved, ensuring the integrity and reliability of our dataset`", which is a number of dialogues filtered out in this process. We believe that this can serve as a substitution for an inter-annotator agreement study.
> >
> > > in the human refinement process, if the subjects choose the distractor/disturbing choice their work is rejected. But couldn't it be the case that the distractor/disturbing choice just happened to be equally good as the original, given the open-ended nature of the task?
> >
> > **A**:  We take the following two operations to improve the annotation quality:
> > 1. We didn't reject **all** of a worker's work once he/she chose a disturbing choice. As mentioned in line 152, we reject the works of those who **frequently** chose disturbing choices.
> > 2. We manually check some of the disturbing choices, and we find out that in most cases, they are totally different from the original choices due to the complexity of context.
> >
> > In the case described, the work is not added into our dataset and a waste of data occurs. However, it's not a big problem, because wasting few of the data during the curation process is inevitable. Furthermore, this is a rare situation and doesn't affect the quality of our dataset.

---

> > > ### Author Response · Authors · 2023-08-24
> > > **Response to Reviewer r1EM (3)**
> > >
> > > > what is the column manually referring to?
> > >
> > > **A**: We follow MuTual [Leyang et al] to use "manually" to refer to "manual annotation", in other words, whether the dataset is being checked or collected by humans.  Thank you for pointing this out. We have added this description to our revised submission.
> > >
> > > > what is the difference between "diverse" and "open" in terms of the domain?
> > >
> > > **A**: Generally, "diverse" means that the dataset lies in several specific domains while "open" means that the dataset doesn't fall into particular domains. Thanks for pointing it out! We have added this clarification to our revised submission.
> > >
> > > > in section 3.1, the "disturbing" choices would be better called "distractor" choices.
> > >
> > > **A**: Thanks for the advice! We have fixed the expression in the revised submission.
> > >
> > > > On line 307, which model performances improve by 38.47%?
> > >
> > > **A**: In line 307, we mentioned that "**On average**, the model performances improve by 38.47% after the introduction of human-annotated rationales.". The numerical result is an averaged performance improvement ratio of all listed models (BART-base, T5-small, mT5-small, UnifiedQA-base, UnifiedQA-large, ChatGPT (em), and ChatGPT (pm) ). Thanks for your question!
> > >
> > >
> > > **Relation to Prior Work**
> > >
> > > > It is unclear what was missing from previous work which calls for this new dataset. Annotated data is always welcome, but what does this enable us to do that was not previously possible?
> > >
> > >
> > > **A**: We highlight the major difference and movitations of our dataset and prior works as follows:
> > >
> > > 1. **New Setting with pragmatic reasoning**: To the best of our knowledge, Diplomat is the first dataset that formally study the problem of pragmatic reasoning under the setting of multi-turn conversations. In the paper, we formally define the problem of pragmatic reasoning, and we offer a carefully curated new dataset. Along with the dataset, we propose two tasks for benchmarking current models, leading to a new approach for the problem of pragmatic reasoning. As acknowedledged by reviewer tasC, the problem is important as well as underexplored, and it's an essential field for advancing the field of Natural Language Processing.
> > >
> > >
> > >
> > > 2. **A need for unified framework**: Previous datasets of literal interpretations of figurative language inspect different aspects of the problem separately, and they mostly lie in different domains as well as containing different tasks. Therefore, with the rising trend of building human-like AI, we believe that a unifed framework is an urgent need.
> > >
> > > 3. **Pursuit of human-like social Language models**:
> > >     - Pragmatic reasoning is a key step towards **general emotional intelligence**. It's common that *same words mean differently under different context settings*, such as "thank you" may express gratitude commonly but it's also possible to be a sacarstic expression under some situations. Understanding such utterances rather than interpreting them literally is essential for building future social chatbots.
> > >     - Pragmatic reasoning is a **fundation of human cognition**, such as intention understanding, theory-of-mind modeling, and value alignment. Our work aims at filling the gap between current LMs and a general emotional intelligent language models that can handle different conversation situations as human beings.
> > >     - From experiments provided in our paper, current powerful LLMs cannot reach satisfying performance of dataset, even after finetuning. Besides providing data corpus for finetuning, our dataset also presents a **valuable evaluation benchmark for LLMs on pragmatic reasoning**.
> > >
> > >
> > >
> > > > Literal interpretations of figurative language and conversational question answering for example have their own, separate datasets. What is the benefit of having them on one?
> > >
> > > We summarize the benefits of having a unified model for figurative language and CQA as follows:
> > >
> > > 1. **Importance of Figurative Language Understanding:** Our main focus of this work is to enhance current LMs with pragmatic reasoning ability, where pragmatics can only be discussed in the context of conversational understanding. Furthermore, understanding figurative language is required in this situation.
> > > 2. **The Need for Unified Modeling:** As discussed before, considering literal interpretation of figurative language unifily is one of the aspects of pragmatic reasoning, which enables the pursuit of a general conversation model. Please refer to response to "Relation to Prior Work" for our detailed clarification.

---

> > > > ### Author Response · Authors · 2023-08-24
> > > > **Response to Reviewer r1EM (4)**
> > > >
> > > > > Previous work, e.g. GRICE and Mutual are critized for their lack of diversity, but then we see that the proposed dataset is about politics and world events, so perhaps the diversity of each dataset should be measured and compared for a fairer assessment.
> > > >
> > > > **A**:  Thanks for pointing this out! We'd like to emphasize 3 aspects:
> > > > 1.  The topic domains of our dataset are open, not all of them are about politics, which can be demonstrated in the examples listed in Figure 2 of our main paper. Furthermore, the 5 reasoning types shown in Figure 2 are also evidences of the large diversity of our dataset.
> > > >
> > > > 2. Furthermore, we'd like to highlight a quantitative feature: vocabulary size (as listed in the table below). This feature can demonstrate the diversity of our work and prove that our dataset holds a satisfying variety.
> > > >
> > > >
> > > > | Dataset|Vocabulary Size|
> > > > |-|-|
> > > > |Diplomat | **48,900**|
> > > > |DREAM| 13,037 |
> > > > |MuTual|11,343|
> > > > |Drop|  29,929|
> > > >
> > > >
> > > >
> > > > 3. Apart from above, the Sunburst chart of Figure 3a and Figure 3b can also prove the variety of Diplomat: Figure 3a shows the words order of pragmatic turns of our dialogues, and Figure 3b shows the words order of questions of our dataset.
> > > >
> > > >
> > > >
> > > >
> > > > > For task 2 I have some concerns on both the question generation and the answers. Why is it useful/interesting/etc. to have systems developed to answers questions that were generated by ChatGPT, as opposed to humans themselves? How do we know these questions are useful?
> > > >
> > > > **A**: Thank you for the valuable question! We'd like to answer the question from 3 aspects:
> > > > 1. **Generated Questions are Reasonable**: It's shown by Lingo et.al. [a] that ChatGPT is feasible when provided with well-crafted prompts. The prompt we use is demonstrated in **Table 6** of the supplementary materials, which by manually checking 200 questions generated, we believe that the questions generated by this pipeline are reasonable. More than that, through the process of humans writing the answers, we haven't received messages telling us that our questions are totally illogical. Apart from that, we conduct a second sampling check (200 question-answer pairs) of answers manually, and we found the data curation satisfying.
> > > > 2. **Generated Questions are more diverse**:  For a good question, it needs to be creative, diverse and related. It's nothing to do with model's reasoning ability. Furthermore, human questions are less diverse and main contain some mistakes. Therefore, we prefer ChatGPT generated questions.
> > > > 3. **Auto-generated question but human annotated answers:** Answers are for evaluating models' ability, reasoning is required and therefore we ask human to offer as well as refine answers.
> > > >
> > > > [a] Ryan Lingo. Exploring the Potential of AI-Generated Synthetic Datasets: A Case Study on Telematics Data with ChatGPT, 2023
> > > >
> > > >
> > > > >  And how did you ensure the quality of the answers by AMT? Quality issues are known, but they have now been exacerbated as crowdworkers now use chatGPT for their work; see here: https://arxiv.org/abs/2306.07899. Again inter-annotator agreement would have been appropriate
> > > >
> > > > **A:**  Thank you for pointing out this issue! We carefully considering avoiding ChatGPT usage ahead of data curation process. We summarize our consideration to be 3 aspects:
> > > > 1. We formally warn the workers that using AI tool is forbidden during our data annotation.
> > > > 2. ChatGPT itself performs poorly on our task (demonstrated in Table 3,5), thus using ChatGPT will lead to logical errors, and inconsistent answers, which will result in rejection of the annotation.
> > > >
> > > > 3. We manually check the appropriateness of data, with the consideration of logic, correctness, and consistency. We will also reject those annotations with explicit AI patterns, for example "As a large language model, I ...", and block those workers from our assignments.
> > > >
> > > >
> > > >
> > > > > Also how did you assess the challenge posed by the questions? You need to compare against different datasets.
> > > >
> > > > **A:** Thanks for asking this question! It's hard to quantitatively evaluate the difficulty of a single question. We list the following metrics to demonstrate the difficulty of the whole Conversational Question Answering task:
> > > > 1. **Question diversity:** In Figure 3b, we have demonstrated the diversity of our questions is considerable.
> > > > 2. **Reasoning Types:** As shown in Figure 2, the process of deriving from questions to answers involves 5 types of reasoning procedure.
> > > > 3. **Question Complexity**: In Table 4, we report the results of several models including the SOTA model (ChatGPT). The results demonstrate that Conversational Question Answering task is challenging for current models.

---

> > > > > ### Author Response · Authors · 2023-08-24
> > > > > **Response to Reviewer r1EM (5)**
> > > > >
> > > > > **Additional Feedback**
> > > > >
> > > > > > Why not ask models to generate rationales in CP->R? It should be within reach of recent large language models.
> > > > >
> > > > > **A**: Thanks for the question!
> > > > > - We want to evaluate the performance of  discriminative (encoder-only) models and generative models, therefore we choose a retrieval-based task. Most previous work, such as VCR[a] formalize this as a retrieval-based task. The benefit is that it's easier to evaluation and is easier for model to perform.
> > > > >
> > > > >
> > > > >
> > > > > - Under the context of Chain-Of-Thought prompting and related works, we agree that bringing explanation generation into a generative model is a valuable suggestion towards the pursuit of reliable generative LLMs. Indeed, we've evaluate such potential on ChatGPT (Table 4). We'll like to investigate more on this topic in the future.
> > > > >
> > > > > [a] Zellers et al. From Recognition to Cognition: Visual Commonsense Reasoning
> > > > > > Why not ask models to identify when pragmatic reasoning is needed in C->R? it would be more realistic
> > > > >
> > > > > **A**: Thanks for asking! Our first task(PIR) includes 3 settings: C->P, CP->R, C->PR. We guess the question is asking C->P. We would like to discuss more if our understanding is incorrect.
> > > > >
> > > > >
> > > > >
> > > > > For C->P task, in line 197, we describe the task of C->P as follows: "`For each instance, models are presented with a dialogue and a specific turn extracted from that dialogue. They are then required to determine whether the given turn qualifies as a pragmatic turn.`" The process of determine the pragmatic turn is **identical** to determining when pragmatic reasoning is needed.
> > > > >
> > > > >
> > > > > For C->PR task, the task consists of 2 steps： C->P, CP->R. The model first needs to identify the pragmatic turn and if its selection is correct, it needs to choose a rationale. The first step requires a pick for pragmatic turn is indeed asking the model when pragmatic reasoning is needed.

---

> > > > > > ### Comment · Reviewer_r1EM · 2023-08-29
> > > > > >
> > > > > > Indeed thanks for the clarifications.

---

> > > > > > > ### Author Response · Authors · 2023-08-29
> > > > > > >
> > > > > > > Thank you for carefully reading our rebuttal and raising your insightful comments! We would like to further clarify our considerations to address your concerns.
> > > > > > >
> > > > > > > > which was not asking to cite papers on the topic of cognitive linguistics in general (which you did), but which ones you used in developing the annotation for the dataset.
> > > > > > >
> > > > > > > **A:** Apologies for misunderstanding the question. The cognitive linguistic theories mainly motivate us to investigate the importance of pragmatic reasoning and commonly occurred pragmatic utterances in daily conversations, which we mainly refer to for choices of reasoning types. Moreover, we refer to recent advances in language reasoning in NLP literature (e.g., [b]) which bring in open-world knowledge reasoning to better facilitate the build of future language models.
> > > > > > >
> > > > > > > [a] Keith Brown, Eve V.Clark, Jim Miller, Lesley Milroy, Geoffrey K. Pullum, and Peter Roach. Meaning in Language: An Introduction to Semantics and Pragmatics, 2004
> > > > > > >
> > > > > > > [b] Wei et. al., Chain of thought prompting elicits reasoning in large language models
> > > > > > >
> > > > > > > > However, the lack of an inter-annotation agreement study is a severe shortcoming; it shouldn't be necessary for everything, but for some at least. Also, the distractor being completely different is not necessarily indicative of poor quality given the open-ended nature of the task. In fact, that's what an inter-annotation agreement study would inform us about.
> > > > > > >
> > > > > > > **A:** Thanks for pointing this out! We **do have** an inter-annotator agreement for our data. We would like to clarify two aspects:
> > > > > > >
> > > > > > > - We've mentioned that in order to ensure the quality of our dataset, we conduct the third step: Human-Refinement (A double-check for the previous data). `In this step, the worker selects the answers from the previous workers' work. Therefore, each choice the third worker chooses represents an agreement between he/she and the other worker. As a result, the number of data left after the third step is how much data two workers agree with each other.` The inter-annotator agreement rate is 71.17\% (4177/5869).
> > > > > > >
> > > > > > > - The distractor choices being different is an interesting empirical observation of us, and it's an answer to the previous question. Sorry about the misunderstanding.
> > > > > > >     - In fact, it doesn't matter whether the distractor/disturbing choice is equally good as the original. As the worker selects a distractor choice, we simply discard the data. It's a **Type I** error(false positive), we aim to promise the correctness of our dataset, which means the correctness of the discarded data set is out of our consideration.
> > > > > > >
> > > > > > > > general vs diverse: It sounds as they mean almost the same thing, not sure the distinction is useful.
> > > > > > >
> > > > > > > **A:** We guess you are concerned about the differences between "general" vs. "open".
> > > > > > > - "**open**" As denoted in (https://symbl.ai/blog/conversation-understanding-open-domain-vs-closed-domain/), an open-domain system typically means it can understand any conversation in any domain.
> > > > > > > - "**diverse**": It refers to the number of topics and domains in this dataset is large. However, it could still be a **close-domain** system.
> > > > > > >
> > > > > > > We hope this can better distinguish the terms.

---

> > > > > > > > ### Author Response · Authors · 2023-08-29
> > > > > > > >
> > > > > > > > > I am afraid I am not convinced about the arguments for using automatically generated questions for the evaluation. The dataset is supposedly about pragmatic reasoning in conversations among humans, so this defeats the purpose, especially as it is used in the only text generation task considered.
> > > > > > > >
> > > > > > > > **A:** Thanks for pointing this out! We'd like to point out that although the dataset is about pragmatic reasoning, the generation of questions within the conversational question answering does **NOT** require an understanding of pragmatics.
> > > > > > > >
> > > > > > > > - ChatGPT is not good at pragmatic reasoning, but it's not a must for it to propose a relevant question. When we are studying knowledge from a class, we don't need to be an expert to ask the teacher a question on the content of the class.
> > > > > > > > -  Let's suppose that ChatGPT proposes the Riemann hypothesis. The question is valuable, whether should we take it as a treasure or discard it because of its proposer as an AI? The question doesn't need to be precise, it needs to be grammatically correct and creative, which is achieved through ChatGPT's generation.
> > > > > > > >
> > > > > > > > We further justify our automated generation process with:
> > > > > > > > - The answer to the questions of Diplomat is written by human beings in order to guarantee the fairness of evaluation.
> > > > > > > > - As seen, ChatGPT performs badly in this task, indicating no data leaking problem in our setting.
> > > > > > > > - There are many **recent advances** mentioning the feasibility of using **ChatGPT** for question generation [a,b,c,d]. Such as Ubani et al. leverage ChatGPT to generate questions for TREC dataset.
> > > > > > > >
> > > > > > > > [a] Ubani et al. ZeroShotDataAug: Generating and Augmenting Training Data with ChatGPT
> > > > > > > >
> > > > > > > > [b] Liu et al. Summary of ChatGPT-Related Research and Perspective Towards the Future of Large Language Models
> > > > > > > >
> > > > > > > > [c]  Ryan Lingo. Exploring the Potential of AI-Generated Synthetic Datasets: A Case Study on Telematics Data with ChatGPT
> > > > > > > >
> > > > > > > > [d] Bahrini et al. ChatGPT: Applications, Opportunities, and Threats
> > > > > > > >
> > > > > > > > > I am not satisfied by the answer on ensuring quality of AMT work, especially since ChatGTP is used a lot. Again, inter-annotator agreement would have helped.
> > > > > > > >
> > > > > > > >
> > > > > > > > **A:** As mentioned before, we have an inter-annotator agreement study. Thank you!
> > > > > > > >
> > > > > > > >
> > > > > > > > We believe our work is essential to the social intelligence and future LLMs. Please check out our responses and let us know if all concerns are addressed.

---

> > > > > > > > ### Comment · Reviewer_r1EM · 2023-08-29
> > > > > > > > **Inter-annotator agreement clarification**
> > > > > > > >
> > > > > > > > The inter-annotator agreement I am asking about is to ask two annotators to perform the same task unaware of each other's work and the check whether they gave the same response, same text, etc. This was not done, just a third worker was asked to check whether they are happy with a response, but this is not the same thing.

---

> > > > > > > > > ### Author Response · Authors · 2023-08-30
> > > > > > > > >
> > > > > > > > > > The inter-annotator agreement I am asking about is to ask two annotators to perform the same task unaware of each other's work and the check whether they gave the same response, same text, etc. This was not done, just a third worker was asked to check whether they are happy with a response, but this is not the same thing.
> > > > > > > > >
> > > > > > > > > **A**: Thanks for your careful review and the patient reading of our rebuttal! Sorry for our misunderstanding. We indeed **have done** the exact inter-annotator agreement as your description above. In step 2, we ask the two workers to `perform the same task unaware of each other's work` and we `check whether they gave the same response`. The inter-annotator agreement rate of the second step is **69.98\%** (the rate at which the two workers gave `the same response` for the same dialogue turn). Step 3 is a **further improvement** of the annotation process, where we ask a third worker to perform the same task as before. The only difference is that we limit the answers space to be the combination of prior answers, distractor answers, and their own answers. The reason is to limit the syntactic variance of expressions of the same meanings.

---

> > > > > ### Comment · Reviewer_r1EM · 2023-08-29
> > > > >
> > > > > - I am afraid I am not convinced on the arguments for using automatically generated questions for the evaluation. The dataset is supposedly about pragmatic reasoning in conversations among humans, so this defeats the purpose, especially as it is used in the only text generation task considered.
> > > > >
> > > > > - I am not satisfied by the answer on ensuring quality of AMT work, especially since ChatGTP is used a lot. Again, inter-annotator agreement would have helped.

---

> > > > ### Comment · Reviewer_r1EM · 2023-08-29
> > > >
> > > > - general vs diverse: it sounds as the mean almost the same thing, not sure the distinction is useful.
> > > > - thanks for the other clarifications/elaborations.

---

> > > ### Comment · Reviewer_r1EM · 2023-08-29
> > >
> > > - I appreciate the inclusion of discussion of Grice's maxims.
> > >
> > > - However, the lack of an inter-annotation agreement study is a serious shortcoming; it shouldn't be necessary for everything, but for some at least. Also, the distractor being completely different is not necessarily indicative of poor quality given the open-ended nature of the task. In fact, that's what an inter-annotation agreement study would inform us about.

---

> > ### Comment · Reviewer_r1EM · 2023-08-29
> >
> > Thanks for your response. However it didn't answer my question, which was not asking to cite papers on the topic of cognitive linguistics in general (which you did), but which ones you used in developing the annotation for the dataset.
> >
> > I accept the arguments offered for treating metaphors, puns and idioms as one category.

---

### Official Review · Reviewer_CPTz · 2023-07-23
**Review for Submission814**

**Rating:** 7
**Confidence:** 4
**Clarity:** The paper is generally well written, …

**Strengths:**

- Generally well-written paper, good flow.
- Pragmatic reasoning is an important ability for conversational agents to have, and is not sufficiently addressed by SOTA models.
- The full dataset and benchmark code is available.

**Additional Feedback:**

None

Update 2023-08-28: The authors have done a nice job updating the paper to incorporate my feedback. They have addressed all of the concerns I brought up in my review. I am updating my score to a 7 (good paper, accept).

**Correctness:**

- The claims in the submission seem to be backed up by the experimental results.
- The dataset seems to be constructed in a sound way.
- The evaluation and experiment design seems appropriate and correct.

**Documentation:**

- There is sufficient detail on data collection, organization, availability, and maintenance.
- There is detail on intended uses.
- There is no documentation on the data format.
- The code to reproduce the benchmarks has been provided as part of the supplementary material.

**Ethics:**

- Ethical question - what does $0.1 per completed task correspond to in hourly wages? How long does each task take? And how does the effective hourly wage compare to the local minimum wage?

**Limitations:**

The authors adequately address the limitations and potential negative societal impact of their work.

**Opportunities For Improvement:**

### Style/grammar/spelling issues

- Line 57: Suggested rewording: owns --> comprises
- Lines 58 - 59: No need for $ $ around the numbers, the typesetting looks strange.
- Line 76: Add a space after GRICE
- Line 97: Typo: "appear" --> "appeared"
- Line 98: Typo: "Pragmatics Reasoning" -->  "Pragmatic reasoning"
- Line 107: No need for the words 'works such as those by'
- Line 108: Typo: delves --> delving
- Line 211: Is 'despite' the right word here? Perhaps 'additionally' would be better.
- Figure 2 has a number of errors:
  - remakrable --> remarkable
  - darest --> darkest
  - Holocause --> Holocaust
  - stifle --> stifling of
  - suggest not --> suggests not
- Table 2
  - Incorrect caption: PunchLine --> Diplomat
  - Insert space between Avg. and Human
- Line 223: Suggested rewording: ability in applying --> ability to apply
- Line 224: What do you mean by 'natural language system'?
- Line 253: Statistical feature --> Statistical features
- Lines 253 - 255: 'These questions ... vary a lot' - run on sentence? Unclear what is meant by 'their following words'.
- Table 3 caption: missing a period at the end of 'percentage'.
- Line 277: There should be a space after (1)
- Lines 279 - 282: Add % signs. Why are there different numbers of digits after the decimal point (90 vs 65.0)?
- Table 3 caption is below table - should be above table according to NeurIPS formatting guidelines.
- Line 363: What do you mean by 'qualified' performances?
- Line 356: 'COT', vs Table 5, 'CoT'
- Line 367: Instead of enclosing 'memorization' in $ signs, use \emph or \textit.
- Line 134: What do you mean by 'from English-speaking countries'?
- Supplementary material
  - Line 6: typo: RoBERT --> RoBERTa
  - Line 19: Suggested rewording: popeq --> popeq implicatures
  - Line 28: well-suited or ill-suited?
  - Figures 1 and 2 are vertically distorted, so they don't look great. It would be better to preserve the original aspect ratio.
  - Line 97: Typo: o assess --> To assess
  - Line 158: Suggested rewording: owns --> comprises
  - Line 183 - what does this phrase mean: "just by giving *a*"?
  - Some of the answers in the datasheet seem incomplete. For example, in lines 337 and 345, the authors simply say 'Yes', without providing the explanation that is asked for in the datasheet template.

### Other issues

- The URL after the title/author block reads `https://punchline-dataset.github.io`. Should it be `https://diplomat-dataset.github.io` instead?
- Line 109: Here, the word 'necessitates' implies the wrong direction in terms of causality -- it implies that the successful completion of the task occurs before the incorporation of commonsense knowledge.
- Line 112: Incorrect section title: 'PunchLine' -> 'Diplomat'
- Figure 5: The question Q2 is a bit ambiguous - at the time Obama made the request, he was president-elect, rather than president. (https://www.politico.com/story/2009/01/obama-and-bush-join-forces-on-bailout-017348) But perhaps you meant 'which former president' instead of 'president', which would disambiguate the context sufficiently.
- Line 215: This seems to conflict with line 149 - what was used to generate the disturbing choices: BERTScore or Sentence-Transformers?

### Checklist
-  The GPU models have been provided, but not how long the training took.

**Relation To Prior Work:**

The authors clearly discuss the relation of this paper to prior work. Table 1, in particular, provides a nice comparison.

**Summary And Contributions:**

This paper introduces a new dialogue dataset (Diplomat) aimed at enabling building conversational agents that are capable of pragmatic reasoning. It also proposes two tasks that are relevant for building agents with human-like reasoning abilities --- pragmatic identification and conversational question answering --- and presents benchmark results on these tasks from SOTA neural models.

---

> ### Author Response · Authors · 2023-08-24
> **Response to Reviewer CPTz (1)**
>
> Thank you for the super careful review and we do appreciate the suggestions on style, grammar, and spelling issues! We have read and modified all raised issues accordingly. Below are our detailed responses.
>
> > Style/grammar/spelling issues
>
> > Line 57: Suggested rewording: owns --> comprises
>
> > Line 76: Add a space after GRICE
>
> > Lines 58 - 59: No need for  around the numbers, the typesetting looks strange.
>
> > Line 97: Typo: "appear" --> "appeared"
>
> > Line 107: No need for the words 'works such as those by'
>
> > Line 108: Typo: delves --> delving
>
> > Line 98: Typo: "Pragmatics Reasoning" --> "Pragmatic Reasoning"
>
> > Figure 2 has a number of errors:
> remakrable --> remarkable
> darest --> darkest
> Holocause --> Holocaust
> stifle --> stifling of
> suggest not --> suggests not
>
> > Table 2
> Incorrect caption: PunchLine --> Diplomat
> Insert space between Avg. and Human
>
> > Line 223: Suggested rewording: ability in applying --> ability to apply
>
> > Line 253: Statistical feature --> Statistical features
>
> > Table 3 caption: missing a period at the end of 'percentage'.
>
> > Line 277: There should be a space after (1)
>
> > Table 3 caption is below table - should be above table according to NeurIPS formatting guidelines.
>
> > Line 356: 'COT', vs Table 5, 'CoT'
>
> > Line 367: Instead of enclosing 'memorization' in $ signs, use \emph or \textit.
>
> > Line 6: typo: RoBERT --> RoBERTa (Supplementary material
> )
>
> > Line 19: Suggested rewording: popeq --> popeq implicatures (Supplementary material
> )
>
> > Line 28: well-suited or ill-suited? (Supplementary material )
>
> > Line 97: Typo: o assess --> To assess (Supplementary material )
>
> > Line 158: Suggested rewording: owns --> comprises (Supplementary material )
>
> > Line 112: Incorrect section title: 'PunchLine' -> 'Diplomat'
>
> > Figure 5: The question Q2 is a bit ambiguous - at the time Obama made the request, he was president-elect, rather than president. (https://www.politico.com/story/2009/01/obama-and-bush-join-forces-on-bailout-017348) But perhaps you meant 'which former president' instead of 'president', which would disambiguate the context sufficiently.
>
>
> Thanks so much for your suggestions, we have corrected these issues in our revised submission.
>
>
>
>
> > Line 211: Is 'despite' the right word here? Perhaps 'additionally' would be better.
>
> Thanks so much for the advice! "Despite" is indeed weird in the context, but we prefer changing into "besides". What do you think of this word?
>
>
>
> > Line 224: What do you mean by 'natural language system'?
>
> We mean the human language system. We have changed the expression of this part to better clarify our intended meaning.
>
>
> > Lines 253 - 255: 'These questions ... vary a lot' - run on sentence? Unclear what is meant by 'their following words'.
>
> In the context, "their following words" represents the words that appear in the question after "What, Which, How". Take a question "Which one is better?" as an example, "one","is" and "better" are following words of "Which". We'll clarify this point better in our revised submission. Thanks for pointing out!
>
>
>
>
>
> > Lines 279 - 282: Add % signs. Why are there different numbers of digits after the decimal point (90 vs 65.0)?
>
> We've added the % signs. The different numbers of digits after the decimal point are typos. Thanks for pointing out!
>
>
>
> > Line 363: What do you mean by 'qualified' performances?
>
> We mean qualified to be a satisfying performance. Thanks for pointing out! We've reword "qualified".
>
>
>
>
>
> > Line 134: What do you mean by 'from English-speaking countries'?
>
> We want our workers to be native speakers, thus we restrict them to coming from one of the English-speaking countries, such as USA, UK and so on.

---

> > ### Author Response · Authors · 2023-08-24
> > **Response to Reviewer CPTz (2)**
> >
> > > Figures 1 and 2 are vertically distorted, so they don't look great. It would be better to preserve the original aspect ratio. (Supplementary material )
> >
> > Thanks for your advice! However, the figure is too long for a single page, and we believe that it may be unconvenient for reading if we distort the figure into two parts.
> >
> >
> >
> > > Line 183 - what does this phrase mean: "just by giving a"? (Supplementary material )
> >
> > In our dataset, errors are inevitable. Here, we mention "a" to be few meaningless annotations only offer an "a" and are included in our dataset by accident. We clean the error cases as we see them, but there are few of them which are not specified in our dataset. However, the error cases are extremely rare, which is a promise that we can make.
> >
> > > Some of the answers in the datasheet seem incomplete. For example, in lines 337 and 345, the authors simply say 'Yes', without providing the explanation that is asked for in the datasheet template.
> >
> > Thanks for pointing out! We'll provide explanation in the revised submission.
> >
> >
> > > The URL after the title/author block reads https://punchline-dataset.github.io. Should it be https://diplomat-dataset.github.io instead?
> >
> > Thanks for pointing out. Both of the urls work.
> >
> > > Line 109: Here, the word 'necessitates' implies the wrong direction in terms of causality -- it implies that the successful completion of the task occurs before the incorporation of commonsense knowledge.
> >
> > Thanks for pointing out! We'll change the word to be "needs".
> >
> >
> >
> >
> > > Line 215: This seems to conflict with line 149 - what was used to generate the disturbing choices: BERTScore or Sentence-Transformers?
> >
> > Thanks for pointing out! In line 149, the disturbing choices generated by leveraging BERTScore is for the human refinement process (A step of data collection). However, the disturbing choices mentioned in line 215 are for constructing the task of our dataset.
> >
> > >Checklist
> > >The GPU models have been provided, but not how long the training took.
> >
> > Thanks for reminding! We have offer a table for training time of each task as well as each model below:
> > | Model | C->P| CP->R| Device|
> > |-|-|-|-|
> > | RoBERTa-large| 2.5min/epoch | 2.8min/epoch | A100|
> > | RoBERTa-base| 0.8min/epoch | 0.9min/epoch | A100|
> > |BERT-base | 0.8min/epoch|0.9min/epoch | A100|
> > | GPT-2 | 5.8min/epoch | 6.2min/epoch | A100|
> > |DialoGPT-medium| 2.4min/epoch| 4.2min/epoch | A100|
> > | DeBERTa-base | 0.9min/epoch|0.9min/epoch | A100|
> > | ALBERT-base | 0.5min/epoch | 0.8min/epoch | A100|
> >
> >
> >
> >
> >
> >
> > > Ethical question - what does $0.1 per completed task correspond to in hourly wages? How long does each task take? And how does the effective hourly wage compare to the local minimum wage?
> >
> > **A:** Thank you for pointing out the ethical concern. We surveyed 10 users to accomplish our task ahead of Turker experiments. All users can complete a single task within 45 seconds, leading to a wage pay of around 8 dollars per hour, which is about a dollar higher than the federal minimum hourly wage of the United States.  Our payment matches the working payload of annotation tasks.
> > We have surveyed Amazon Turk market price of similar tasks (such as reading comprehension, question answering, and language generation...), and the average pay for each assignment is around 0.08 dollars, thus we chose 0.1 dollars per assignment and strict qualification requirements (Refer to Table 7 of Appendix B) to ensure the quality of annotation.

---

> > > ### Comment · Reviewer_CPTz · 2023-08-27
> > > **Response to authors**
> > >
> > > > we have corrected these issues in our revised submission.
> > >
> > > Thanks! However, some of the issues have not been corrected:
> > > - Figure 2 still has the typos
> > > - Figure 5 still has the ambiguity about president vs president-elect.
> > > - Typo on line 6 of supplementary material (RoBERT instead of RoBERTa) is still not fixed.
> > >
> > > > In the context, "their following words" represents the words that appear in the question after "What, Which, How". Take a question "Which one is better?" as an example, "one","is" and "better" are following words of "Which". We'll clarify this point better in our revised submission. Thanks for pointing out!
> > >
> > > Thanks for clarifying. I would suggest the following rewording: "the following words after the interrogative ones" --> "the words that follow the interrogative ones"
> > >
> > > > We want our workers to be native speakers, thus we restrict them to coming from one of the English-speaking countries, such as USA, UK and so on.
> > >
> > > I strongly recommend listing the countries explicitly in the supplementary material.
> > >
> > > > Thanks for your advice! However, the figure is too long for a single page, and we believe that it may be unconvenient for reading if we distort the figure into two parts.
> > >
> > > Ok, sounds good.
> > >
> > > > Here, we mention "a" to be few meaningless annotations only offer an "a" and are included in our dataset by accident.
> > >
> > > Apologies, I am still not clear what is meant by 'offer an "a"'.  Could you give an example? It would also be good to expand the explanation in the supplementary material itself.
> > >
> > > > We'll provide explanation in the revised submission
> > >
> > > I don't see the explanations in the revised submission. Is this something you were planning on doing for a future version?
> > >
> > > > We have offer a table for training time of each task as well as each model
> > >
> > > Thank you for providing this info! However, I think this information should be included in the supplementary material or in the checklist itself (I do not see it in the revised submission).
> > >
> > > Re: Question about effective hourly wage --- thank you for the explanation. This information should be included in the supplementary material.
> > >
> > > > strict qualification requirements (Refer to Table 7 of Appendix B) to ensure the quality of annotation.
> > >
> > > The caption for Table 7 reads: "Hyperparameters For Models on CQA." Are you referring to a different table perhaps? If so, which one?
> > >
> > > New typos detected
> > > - Line 134: "redundancy, thus" --> "redundancy, and thus"

---

> > > > ### Author Response · Authors · 2023-08-28
> > > >
> > > > Thanks so much for the careful review and additional comments. We have addressed all additional feedback and are happy to hear your further comments.
> > > > > Figure 2 still has the typos
> > > > > Figure 5 still has the ambiguity about president vs president-elect.
> > > >
> > > > We want to point out that some typos are **Dialogue Context** typos, which come from  the source dataset (transcripts of TV shows), therefore we tend to use the original data to get close to real-life corpus. We have fixed ones in Figures per your comments.
> > > >
> > > > > Typo on line 6 of supplementary material (RoBERT instead of RoBERTa) is still not fixed.
> > > >
> > > > Sorry we just noticed it is allowed to submit a revised supplementary material,. We've just uploaded a new version of suplementary materials.
> > > >
> > > > > Thanks for clarifying. I would suggest the following rewording: "the following words after the interrogative ones" --> "the words that follow the interrogative ones"
> > > >
> > > > We agree! Thanks so much, we have reworded it.
> > > >
> > > > > We want our workers to be native speakers, thus we restrict them to coming from one of the English-speaking countries, such as USA, UK and so on.
> > > >
> > > > This feature is listed in Table 7 of the supplementary material (revised submission). Thanks for pointing out!
> > > >
> > > > > Here, we mention "a" to be few meaningless annotations only offer an "a" and are included in our dataset by accident.
> > > > > Apologies, I am still not clear what is meant by 'offer an "a"'. Could you give an example? It would also be good to expand the explanation in the supplementary material itself.
> > > >
> > > >
> > > > Sorry for our misclarification. Some workers may want to finish the work as quickly as possible, therefore when we ask them to offer a rationale for choosing a certain turn as a pragmatic turn, they simply type an "a" in the box. However, the situation is rare, and we blocked the workers and clean the data out of our dataset. Thanks for pointing out, we have clarified it in the revised submission.
> > > >
> > > > > Some of the answers in the datasheet seem incomplete. For example, in lines 337 and 345, the authors simply say 'Yes', without providing the explanation that is asked for in the datasheet template.
> > > >
> > > > We have clarified this in our updated supplementary materials, thanks so much!
> > > >
> > > > > We have offer a table for training time of each task as well as each model
> > > > > Thank you for providing this info! However, I think this information should be included in the supplementary material or in the checklist itself (I do not see it in the revised submission).
> > > >
> > > > Thanks so much, we've added it to the updated supplementary materials.
> > > >
> > > > > Re: Question about effective hourly wage --- thank you for the explanation. This information should be included in the supplementary material.
> > > >
> > > >
> > > > Thanks for pointing out! We've added it to the updated supplementary materials.
> > > >
> > > > > Strict qualification requirements (Refer to Table 7 of Appendix B) to ensure the quality of annotation.
> > > > > The caption for Table 7 reads: "Hyperparameters For Models on CQA." Are you referring to a different table perhaps? If so, which one?
> > > >
> > > > Sorry for the wrong reference. We refer to Table 7 of the newly updated supplementary material, which is a table on AMT workers requirements.
> > > >
> > > > > New typos detected
> > > >
> > > > > Line 134: "redundancy, thus" --> "redundancy, and thus"
> > > >
> > > > Thanks so much for pointing out, we've corrected it!

---

> > > > > ### Comment · Reviewer_CPTz · 2023-08-28
> > > > > **Updating score**
> > > > >
> > > > > The authors have done a nice job addressing my concerns. I am increasing my score to 7 (good paper, accept). I am hesitant to give a higher score, since this is my first time reviewing for NeurIPS, and I don't know if I have read enough NeurIPS papers to be able to judge whether the paper lies in the top X% of accepted papers.

---

> > > > > > ### Author Response · Authors · 2023-08-30
> > > > > >
> > > > > > Thanks so much for your careful review!

---

### Official Review · Reviewer_xhK2 · 2023-07-28
**Valuable ideas and dataset but rather small and not diverse enough**

**Rating:** 6
**Confidence:** 2
**Clarity:** Yes, the paper is well written.

**Strengths:**

The main strength of the submission is sharing with the broader research community a new valuable, open and free dataset that was created by fairly paying annotators at AMT.
Another strong suit is drawing the research community's attention to the importance of pragmatic reasoning in tasks that deal with dialogues and successfully demonstrating this gap in experiments.
Submission with primarily supplementary material proves beyond the shadow of a doubt the high quality of the research.
The ethical and social implications of a paper are positive. The research community will gain new datasets to experiment with and materials build upon.


**Additional Feedback:**

1. Are there more reasons for the results of RoBERTa large (60.8) being much worse than of RoBERTa base (92.0) on PIR  CP->R as well as the result of RoBERTA large being the worst in PIR C -> PR with the lowest result 0? - per supplementary materials page 2 table 1.

2. Is there a discrepancy between the approval rate / acceptance threshold in supplementary materials (95%) and in the main paper (98%)?


**Correctness:**

Yes - claims are correct and the Diplomat dataset is constructed in a sound way.

**Documentation:**

The dataset is well documented and all required components are included.

**Ethics:**

Not really, I don’t.

**Limitations:**

There are no negative societal impacts, as for limitations authors identified focus on politics and absence of subjectivity and discussed them. A possible way of addressing these limitations is included in the opportunities for improvement section but in short larger and more diverse dataset is the main thing that could help.

**Opportunities For Improvement:**

The main opportunity for improvement is to process by the same pipeline and conduct the same experiments but with much more dialogue. That would help to create a more diverse dataset, make experiment results more robust and enable analysis with regard to various topics and categories, and open up the research community for more widespread adaptation of this work.


**Relation To Prior Work:**

Yes, a description of what makes this work distinct is conveyed in the paper.

**Summary And Contributions:**

Submission introduces a new benchmark and pragmatic reasoning in the conversations dataset (4117 dialogues created using AMT).
Moreover, it proposes two tasks, namely pragmatic identification and reasoning and conversational question answering.
Experiments with SOTA models, including renowned LLMs, demonstrate various performance and quality issues that authors attribute to a lack of context understanding and pragmatic awareness,

---

> ### Author Response · Authors · 2023-08-24
> **Response to Reviewer xhK2**
>
> Thanks for your review and help suggestions! We are delighted to have further and deeper discussion, so that we can gain more valuable insights on the development of Pragmatic Reasoning.
>
> > The main opportunity for improvement is to process by the same pipeline and conduct the same experiments but with much more dialogue. That would help to create a more diverse dataset, make experiment results more robust and enable analysis with regard to various topics and categories, and open up the research community for more widespread adaptation of this work.
>
> **A:** Thanks so much for your suggestion! We would like to point out that the Diplomat possesses:
> - **Considerable Diversity**: In Table 2, our dataset has a vocabulary size of 48,900 (11,343 in MuTual and XXX in ScienceQA). As shown in Figure 3 (the Sunburst figure),
> - **Well-curated Quality:** The large diversity of our dataset ensures the difficulties of the proposed tasks, and they are extremely challenging for current models, including SOTA LLMs. It's worth mentioning that the collection process of pragmatic data is challenging.
> - **Data Collecting is Difficult:** It's extremely difficult to collect considerable amount of pragmatic data. Reasons are as follows:
>     - 1. It's hard to automatically capture pragmatic data from daily dialogues. The process requires commonsense, culture, and proficiency in language, thus we recruit native speakers to do the job
>     - 2. Although pragmatic phenomenon occurs frequently and pragmatic data is widely spread in daily conversations, they holds diversed pattern and therefore it's impossible to summarize a general rule for spotting them.
>
>
> **How will we enhance the Diplomat dataset?** In the future, we'll leverage the pipeline for curating more data, and we may consider data from other sources including Talk Show, Friends, Taxi and so on. Apart from that, we'll improve our dataset according to feedbacks from other users.
>
>
> > 1. Are there more reasons for the results of RoBERTa large (60.8) being much worse than of RoBERTa base (92.0) on PIR CP->R as well as the result of RoBERTA large being the worst in PIR C -> PR with the lowest result 0? - per supplementary materials page 2 table 1.
>
> **A:** We appreciate your careful review. This is indeed an interesting discovery of our experiments: In general, a large model is expected to perform better than a base model. However, we got a result contradicting our expectation. We hypothesize the reasons to be two-fold:
> 1. **domain discrepancy between pragmatic data and LM's pretraining data**: Larger models possess a stable effect of pretraining, while smaller models are easy to shift during finetuning. Of note, beyond using semantic knowledging, our pragmatic data yields deeper reasoning challenge to LMs, thus smaller models can "superfacially" memorize the pattern within the finetuning data, whereas large LMs stick to their original understanding. Similar observations are also shown in multi-step Math Reasoning [a] and few-shot learning [b].
> 2. **Low-resource and diversity of pragmatic data:** Pragmatic data are scarce and diverse.  It essentially requires model go beyond semantic pattern memorizing but to reasoning over the context, social commonsense, and even theory-of-minds. Such low-resource setting contracts the conventional finetuning condition of large LMs that requires massive data to achieve relatively good performance.
>
> [a] Fu et al., Specializing Smaller Language Models towards Multi-Step Reasoning
> [b] Schick et al., It’s Not Just Size That Matters:
> Small Language Models Are Also Few-Shot Learners
>
> To offer a evidence for our hypotheses, we conduct a zero-shot version of Task 1(PIR). The results are reported in the following table:
>  | Model  | $C \rightarrow P$| $CP \rightarrow R$ | $C \rightarrow PR$|
> |--- | --- | --- | --- |
> | $RoBERTa_{large}$ (zero-shot) | 61.1 | 21.8 | 1.7|
> | $RoBERTa_{base}$ (zero-shot) | 57.16 | 21.47 | 5.08|
> | $RoBERTa_{large}$ (original-paper) | 63.8 $\pm$ 0.0 | 60.8 $\pm$ 0.5 | 0.0 $\pm$ 0.0 |
> | $RoBERTa_{base}$ (original-paper) | 64.4 $\pm$ 1.3 | 92.0 $\pm$ 0.4 | 50.0 $\pm$ 11.28|
>
> Comparing the result of zero-shot setting and results reported in the original paper, we can easily observe that the improvement of $RoBERTa_{base}$ after finetuning is much larger than that of $RoBERTa_{large}$. This result offer an empirical proof for our hypotheses.
>
> > 2. Is there a discrepancy between the approval rate / acceptance threshold in supplementary materials (95%) and in the main paper (98%)?
>
> **A:** 95% that is mentioned in the supplementary materials corresponds to **Step 3** of our data curation process: Human Refinement (a double check for dataset), and 98% mentioned in the main paper corresponds to **Step 2** of our collection process: Fine-grained Annotation. Thank you for the question. We have clarified this in the revision.

---

### Author Response · Authors · 2023-08-24
**Response to all reviewers**

We appreciate efforts of all reviewers and thanks for their wonderful suggestions!
We highlight main strengths and contributions of our work as follows:

- **Valuable and Useful Dataset**: We derive "*a new valuable, open and free dataset*" (Reviewer xhK2) that can be shared with the research community. More than that, our dataset "*demonstrates its robustness through rigorous filtering, showcasing a wide array of topics and rich diversity sourced from real-life corpora*" (Reviewer tasC). By "*focusing on dialogues from real-world interviews*"(Reviewer r1EM) and conducting several effective measures, our work is diverse and is "*beyond the shadow of a doubt the high quality research*"(Reviewer xhK2).
- **Important and Novel Problem Setting**: We "address the problem of Pragmatic Reasoning"(Reviewer tasC) and highlight "*the importance of pragmatic reasoning*" (Reviewer xhK2, Reviewer tasC) by "*demonstrating the gap in experiment*" (Reviewer xhK2), "highlighting the limitations of current models in understanding pragmatic awareness." (Reviewer hCbW) as well as proving the problem "*is not sufficiently addressed by SOTA models*" (Reviewer CPTz). We " offer valuable insights that can guide future research in the field of pragmatic reasoning." (Reviewer hCbW)
- **Broad Impact**: We "*highlight the limitations of current powerful language models, such as ChatGPT,*" (Reviewer tasC) and in this aspect, "*our work paves the way for further advancements*" (Reviewer tasC)

We've revised our manuscript per the reviewers' suggestions (highlighted in red in the uploaded revision pdf). Detailed responses to each reviewer's concerns are carefully addressed point-by-point. Below summarize the major updates we've made:
- We clarify certain words in **Table 1**
- A description of the relation between Grice maxims and our work. (**Section 2: Related Work**)
- A figure describing models' ability in different dimensions of Pragmatic Reasoning. (**Figure 6**)
- A description of the relation between Zero-Shot NLI and the rest of our proposed tasks. (**Section 5.3**)

---

### Decision · Program_Chairs · 2023-09-22

**Decision:**

Accept (Poster)

**Comment:**

This paper presents a dataset designed to measure pragmatic reasoning in models, an interesting challenge area for current language models.  Many weaknesses from reviewers were addressed and this should be an interesting addition to the Benchmarks track.